# Maturational networks of human fetal brain activity reveal emerging connectivity patterns prior to ex-utero exposure

Vyacheslav R. Karolis [1,2✉], Sean P. Fitzgibbon[2], Lucilio Cordero-Grande [3], Seyedeh-Rezvan Farahibozorg[2], Anthony N. Price[1], Emer J. Hughes[1], Ahmed E. Fetit[4,5], Vanessa Kyriakopoulou [1], Maximilian Pietsch[1], Mary A. Rutherford[1], Daniel Rueckert [4,6], Joseph V. Hajnal[1], A. David Edwards [1,7], Jonathan O'Muircheartaigh [1,7,8], Eugene P. Duff[2,9,11] & Tomoki Arichi [1,7,10,11]

A key feature of the fetal period is the rapid emergence of organised patterns of spontaneous brain activity. However, characterising this process in utero using functional MRI is inherently challenging and requires analytical methods which can capture the constituent developmental transformations. Here, we introduce a novel analytical framework, termed "maturational networks" (matnets), that achieves this by modelling functional networks as an emerging property of the developing brain. Compared to standard network analysis methods that assume consistent patterns of connectivity across development, our method incorporates age-related changes in connectivity directly into network estimation. We test its performance in a large neonatal sample, finding that the matnets approach characterises adult-like features of functional network architecture with a greater specificity than a standard group-ICA approach; for example, our approach is able to identify a nearly complete default mode network. In the in-utero brain, matnets enables us to reveal the richness of emerging functional connections and the hierarchy of their maturational relationships with remarkable anatomical specificity. We show that the associative areas play a central role within prenatal functional architecture, therefore indicating that functional connections of high-level associative areas start emerging prior to exposure to the extra-utero environment.

[1] Centre for the Developing Brain, School of Biomedical Engineering and Imaging Sciences, King's College London, London, UK. [2] Wellcome Centre for Integrative Neuroimaging, FMRIB, Nuffield Department of Clinical Neurosciences, University of Oxford, Oxford, UK. [3] Biomedical Image Technologies, ETSI Telecomunicación, Universidad Politécnica de Madrid & CIBER-BBN, Madrid, Spain. [4] Biomedical Image Analysis Group, Department of Computing, Imperial College London, London, UK. [5] UKRI CDT in Artificial Intelligence for Healthcare, Department of Computing, Imperial College London, London, UK. [6] Klinikum rechts der Isar, Technical University of Munich, Munich, Germany. [7] MRC Centre for Neurodevelopmental Disorders, King's College London, London, UK. [8] Department of Forensic and Neurodevelopmental Sciences, Institute of Psychiatry, Psychology & Neuroscience, King's College London, London, UK. [9] Department of Brain Sciences, Faculty of Medicine, Imperial College London, London, UK. [10] Paediatric Neurosciences, Evelina London Children's Hospital, Guy's and St Thomas' NHS Foundation Trust, London, UK. [11] These authors contributed equally: Eugene P. Duff, Tomoki Arichi. ✉email: slava.karolis@kcl.ac.uk

Does a 'thing' possess invariant properties that define its 'being', or does its essence reveal itself in the process of a perpetual change, i.e., in its 'becoming'? This ancient intellectual dilemma, conceived by an early Greek philosopher Heraclitus, has been entwined in the centuries-long evolution of human knowledge[1,2]. At its core, it reflects a fundamental problem of selecting an appropriate representational framework for studying a phenomenon while offering a choice between two extreme alternatives. On the one hand, a description of invariant (canonical, typical) characteristics serves a purpose of giving a phenomenon a concrete definition and thus differentiating it from other things. On the other hand, representations that characterise a phenomenon as a process are more fitting if the phenomenon constitutes a sequence of superseding transient states with ill-defined invariant characteristics.

The notion of functional networks in the fetal brain is a case in point for the latter. Evidence from animal models suggests that intrinsically generated neural activity in the prenatal brain first begins with local direct propagation before progressing to larger bursts of spontaneous activity which help to establish local circuitry[3]. At around 26 weeks of gestation, ex-utero functional MRI (fMRI) studies of very preterm infants[4] show that spatially distinct resting-state networks can be identified, initially consisting of local patterns of connectivity with a lack of long range interhemispheric or dorsocaudal connections. Towards term equivalent age, these networks evolve into a set of spatially distributed (multi-nodal) co-activation patterns resembling those seen in adults[5,6], reflecting a generic drift of organic functions towards forming increasingly complex systems[7]. Such rapid developmental changes mean that functional networks in the prenatal period possess the attributes of an intrinsically non-static entity, a characteristic example of Heraclitian "becoming".

Previous research has demonstrated that, despite enormous technological challenges, functional connectivity in utero can also be studied using resting-state fMRI[8–12]. This opens up an opportunity for the use of standard approaches to group-level fMRI network analyses[13] such as group independent component analysis (group-ICA)[14–16]. The latter describes functional networks as a collection of spatial maps[17], each of them charting areas linked together by the strength of covariation between the timecourses of their fluctuating intrinsic activity. However, utility of this method for application with fetal data remains an open question, both conceptually and when considering the unique signal properties of the data acquired in utero. Conceptually, an assumption embedded into this method is that a group-level spatial map characterises a canonical form of a functional network with respect to its individual manifestations, thereby downgrading developmental changes in its spatial layout to the status of non-systematic, and likely underestimated[18], inter-subject variability. On a practical level, application of group-ICA to fetal data typically renders maps of poorly localised and segregated regions, lacking network-like features, such as the presence of spatially non-contingent brain areas[13]. This may be explained by the weakness of long-distance connectivity in the fetal brain but may also be a consequence of inherently high levels of motion and low signal-to-noise ratio in this data, which adversely affects the detection of long-distance connections[19,20]. As a result, coherent developmental features that are fundamental to both a definition and understanding of the neuroscientific basis of functional networks in utero are likely lost using this standard approach.

In this study, we hypothesised that a biologically-motivated analytical framework, that conceptualises functional brain network connectivity as a formative process, may provide a superior modelling alternative to the group-ICA for in-utero data. To this end, to capture the maturational transiency of connectivity states, we introduce an alternative perspective on resting-state functional networks, which we call "maturational networks", or matnets for conciseness. The key feature of this framework is that it incorporates age-related changes in connectivity into network estimation, thereby characterising functional networks as an emerging property of the brain. At its core, it builds on Flechsig's idea[21], that functionally related areas mature together. In contrast to the standard analytical approach of ICA, which utilises correlational structure to factorise networks, our approach leverages age-related changes in correlations in order to characterise maturational modes of variation in the data. The utility of this approach is demonstrated in in-utero fMRI data acquired as part of the developing Human Connectome Project (dHCP)[22,23], an open science initiative aiming to map brain connectivity across the perinatal period, that were reconstructed and preprocessed using specially developed methodologies[24–26]. We show that our approach overcomes inherent limitations of fMRI data acquired in-utero for characterising mid- and long-distance connectivity, and for inference about the developmental trajectory of the fetal functional connectome. Moreover, it enables factorisation of spatial patterns that fit better the concept of resting-state network as we understand it from the studies of more mature brains, that is, as spatially distributed configurations encompassing non-adjacent brain areas[27,28]. Finally, we show that maturational networks lead to new perspectives on the macro-scale developmental relationships in the human brain, the "maturational connectome" and "maturational hubs".

## Results

Resting state fMRI data from 144 healthy fetuses with an age range between 25 and 38 weeks gestation (Supplementary Fig. 1) were acquired over 12.8 mins on a 3 T Philips Achieva system (Best, NL)[29] as part of the developing Human Connectome Project (dHCP). All of the fetal brain images were clinically reported and showed appropriate appearances for their gestational age with no acquired lesions or congenital malformations. The data underwent dynamic geometric correction for distortions, slice-to-volume motion correction[24,25] and temporal denoising[26], followed by their registration to a common space to enable group-level analyses[6].

**The framework**. In order to demonstrate the utility of our approach, we note that developmental changes in a spatial layout of functional networks can be modelled retrospectively within the standard group-ICA approach using several post-processing steps[16], as shown in Fig. 1a. The results of this modelling can therefore serve as a reference for comparison with the results of matnets modelling. In brief, the conventional modelling approach involves the estimation of group-level ("canonical") spatial maps, followed by the two steps of dual regression (DR)[16], i.e., a sequence of spatial and temporal regressions performed against individual data, in order to obtain subject-specific variants of the group maps, followed by a mass-univariate (i.e., voxelwise) modelling of the latter using age as a covariate. The key step is the dual regression step, that "permits the identification of between-subject differences in resting functional connectivity based on between-subject similarities"[16], where a subject-specific map represents the individualised manifestation of a group map.

In contrast, our matnets approach, shown in Fig. 1b, attempts to derive maps of maturational modes of variation in a direct manner, in essence by reversing the order of operations while omitting the intermediate steps of dual regression; that is, we aim to derive spatial maps which themselves are the manifestations of age-related changes in functional connectivity. It runs as follows. At the first step, a dense N voxels by N voxels connectome is

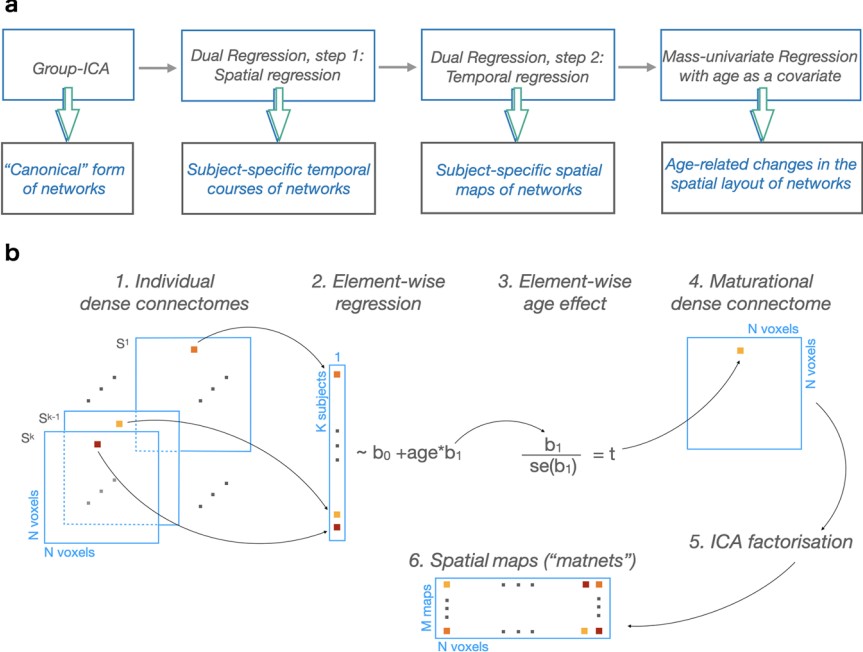

**Fig. 1 Two approaches to maturational analysis of the functional networks. a** Group-ICA + dual regression pipeline and its outputs. The pipeline allows modelling maturational changes in the spatial layout of the networks using mass-univariate analysis of the subject-specific variants of the group maps. The latter are derived using dual regression. **b** Pipeline for derivation of maturational networks. It directly leverages age-related changes to derive networks instead of estimating subject-specific variants of the group-level maps. In the current study: M = 25 (Ref. [6]), N = 53443, K = 144; se - standard error.

computed for each subject separately. Each element of the dense connectome is then fitted across subjects with age as covariate and converted using t-statistics into a maturational dense connectome, i.e., a matrix in which elements contain the estimates of the age effect. An ICA factorisation of the maturational dense connectome is then performed to obtain spatially independent matnet maps, each of them associated with a characteristic profile of emerging connectivity. In other words, as much as temporal correlations between voxels determines their participation in a particular group-ICA network, similarity in the age-related changes in connectivity between voxels determines their matnet participation.

**Univariate spatial properties of group-average correlations and age-related differences in correlations.** The efficiency of either method for network analysis, for instance in terms of their ability to discover meaningful spatial relationships, is contingent on the relevant signal properties of the data, which remain poorly understood for the in-utero fMRI. A brief description of these properties would assist subsequent interpretations and inform analytical choices. Consequently, we provide a short summary of the univariate spatial properties of the two metrics that are expected to shape the results of the group-ICA and matnets analyses: respectively, group-average correlations and the effect of age (t-value) on the strength of correlations.

The generic spatial structure of the two metrics can be easily appreciated by considering connectivity maps from seed regions to the whole brain ("seed-to-brain" maps). The maps of the group-average correlation for six cortical seeds (3 for each hemisphere; estimated from the correlation between the mean timecourse of voxels within a seed mask and timecourses of all voxels in the brain, and then averaged across subjects) are shown in Fig. 2a (left panel). The conspicuous feature of these maps is a presence of a strong distance dependent gradient, indicating signal smearing over the immediate neighbourhood of the seed. This effect transgresses anatomical boundaries, as demonstrated in a context

of interhemispheric connectivity between homologous left and right voxels where the anatomical and purely spatial distances can be disentangled (Supplementary Fig. 2) and shows a spatially indiscriminate character as it could equally be replicated for seeds located in the white matter (Supplementary Fig. 3).

In comparison, the configuration of the spatial maps for the age-related effect on correlation (that is, instead of being averaged across subjects, the seed-to-brain correlation maps were fitted voxel-wise with age as a covariate) for the same set of seeds reveals two components of relevance: a negative local component and a positive mid- and long-distance component (Fig. 2a, right panel). The negative local component is revealed by a distribution of high negative values in the proximity of the seed. This local component, which implies that the strength of distance-dependent gradients in connectivity structure is negatively associated with age at a short distance, occurs in a spatially indiscriminate manner, though less obviously in white matter (Supplementary Fig. 4), possibly due to a greater signal blurring within this tissue. Otherwise, the positive mid- and long-distance component is characterised by an age-related increase in correlation strength between seed and other grey matter regions.

Furthermore, Fig. 2b shows the relationship between the spatial distance and the similarity (i.e., spatial correlation) between 44850 pairs of seed-to-brain maps, computed following the parcellation of the cortex into 300 clusters. The relationship was strong for group-average correlation maps ($r = -0.80$), which suggests that spatial distance may become a dominant factor for the fusion of the voxels into networks in analyses based on the correlational structure of the data, such as group-ICA. Conversely, the similarity between age-effect maps was more robust to the effect of distance between seeds used to produce these maps ($r = -0.42$). This suggests that leveraging positive age-related associations for the network construction can potentially reveal a rich set of spatially distributed patterns with improved specificity. In this view, matnets were derived using a factorisation of the positively thresholded maturational dense connectome.

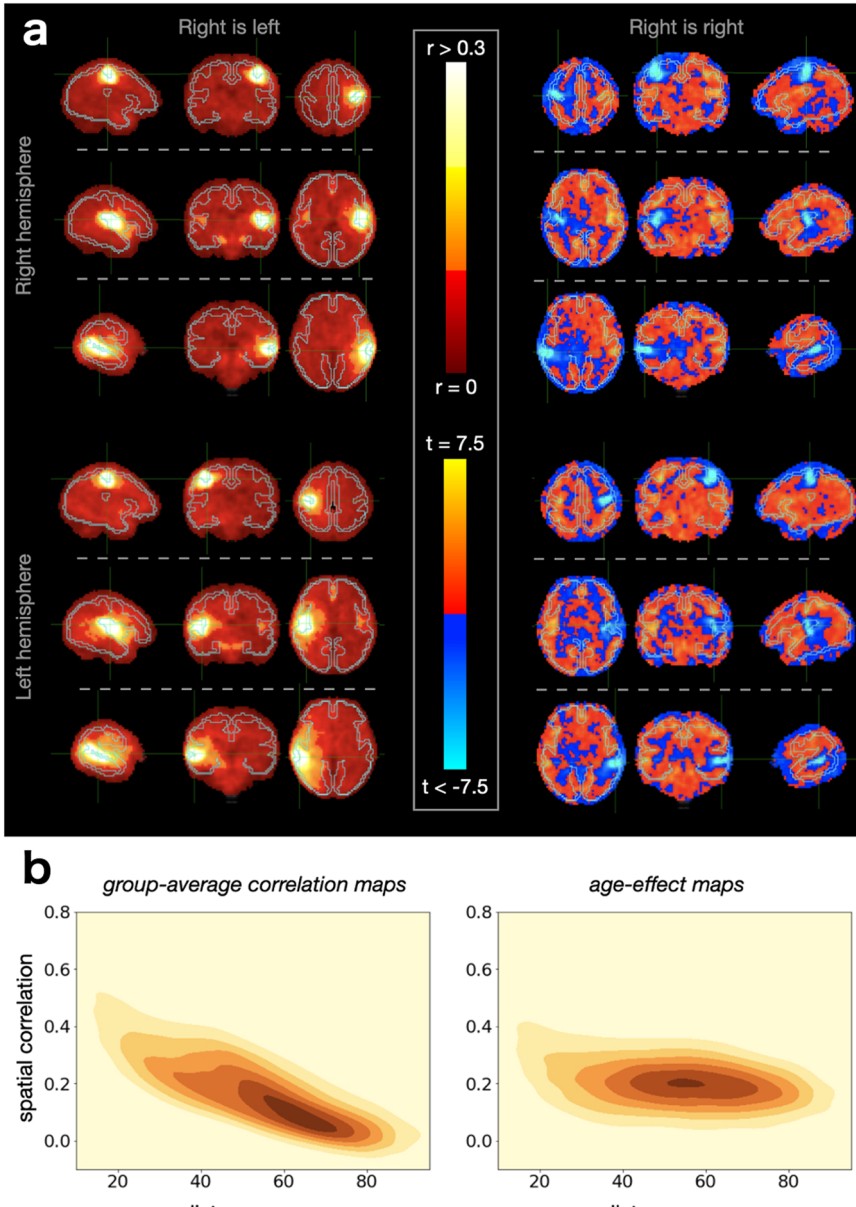

**Fig. 2 Spatial properties of group-average correlations and age-related differences in correlations. a** seed-to-brain maps of group-average correlations (left) and it age-related changes (right). The two types of maps are shown as a mirror-like reflection of each other. Examples of 6 seeds are shown, 3 for each. **b** Distance vs spatial similarity relationship for pairs of seed-to-brain maps.

**Comparison of group-ICA and matnets in neonatal sample.** We first present evaluation of the performance of the matnet framework in the neonatal sample, where the standard approaches proved to be effective and hence meaningful comparisons can be made. For this we constructed a sample of 311 neonates (Supplementary Fig. 5) and obtained group-ICA and matnet factorisations, Fig. 3a, b, respectively. The two methods show an excellent agreement with each other, oftentimes replicating not only the main network nodes but also agreeing on secondary clusters composed of a smaller number of voxels. The following differences can be distinguished qualitatively.

Firstly, in several cases matnets revealed more left-right symmetrical maps than group ICA. The list includes: a bilateral auditory network (matnet #12) compared to its predominantly left- and right-lateralised group-ICA counterparts (gica #6 and gica #8); matnet #6 (occipital pole) compared to gica #13 (right hemisphere dominance) and gica #16 (left hemisphere

dominance); matnet #1 that for group-ICA fractionates into 3 - predominantly medial (#0), predominantly right lateralised (#2) and predominantly left lateralised (#3) - components; a bilateral fronto-parietal matnet #2 (inferior parietal cortex + prefrontal + inferior temporal cortex), that combines areas delineated using 3 group-ICA components, left-dominant gica #12, right-dominant gica #9 and bilateral prefrontal gica #20.

Secondly, matnets provided two non-cortical components, one in the cerebellum (#18) and the other in the brainstem extending into cerebellum (#8). A group-ICA component (#22), spatially similar to the latter, appears to be dominated by the signal originating in CSF and is unlikely to represent an exact match to its matnet counterpart.

Thirdly, matnet #10 provides the most complete delineation of the default mode network in neonates, encompassing all of its critical nodes, including a small cluster in the posterior medial parietal cortex. These regions were contained within two

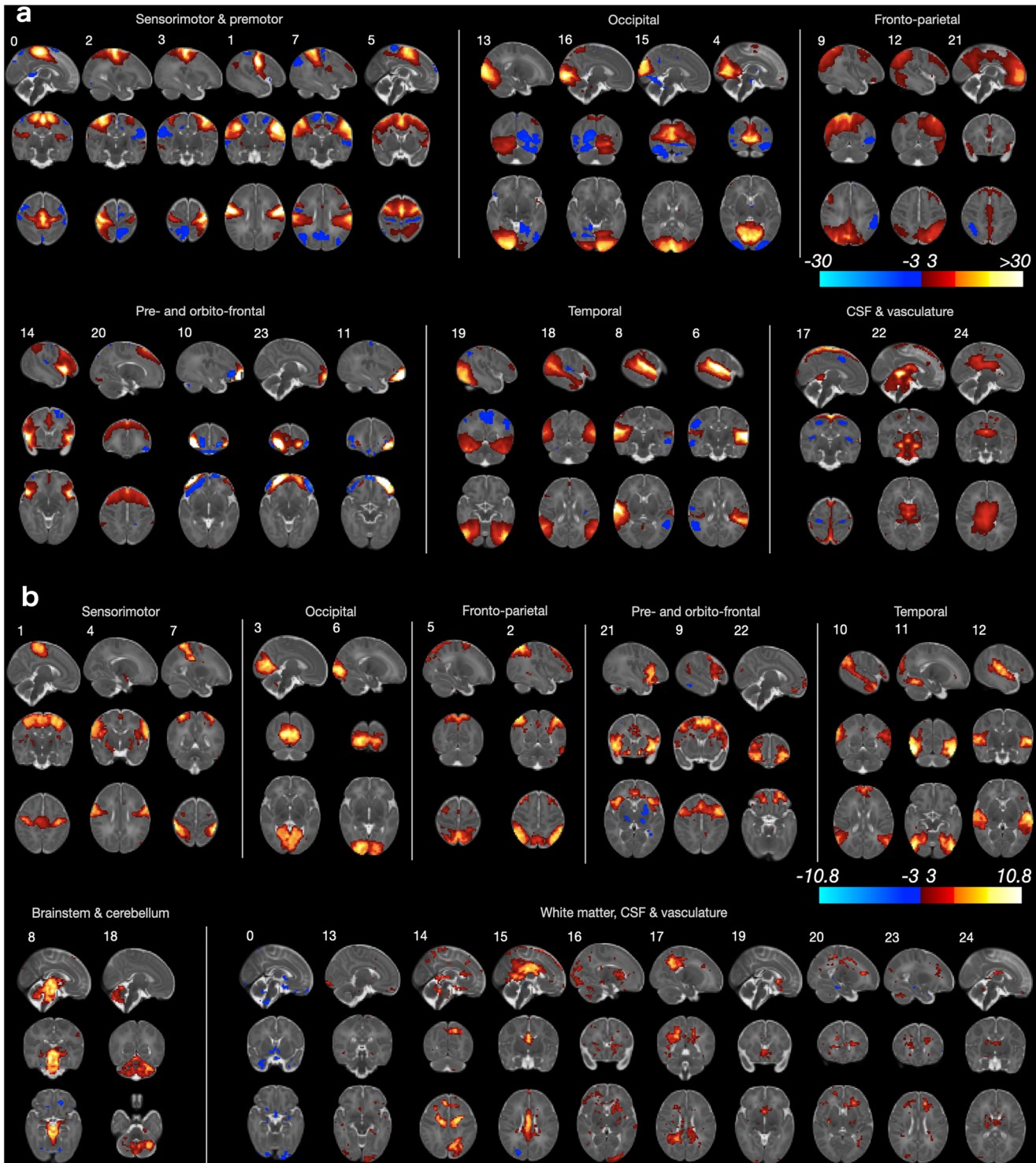

**Fig. 3 Group-level network analyses in neonates.** All spatial maps are shown in radiological orientation. **a** Group-ICA. **b** Matnets.

group-ICA components (#18 and #21), one of which (#21) is likely to be contaminated by the signal originating in the CSF and/or vasculature.

Finally, there was no exact match among matnets to gica #15 (superior medial occipital) and two pairs of matnets-gica components differed on the localisation of their nodes. A prefrontal matnet #9 is shifted anteriorly compared to gica #5 and lacks its posterior node; the (secondary) frontal nodes of predominantly superior parietal matnet #5 are located dorsally in superior frontal gyrus, anteriorly to pre-central sulcus

(supplementary motor area), whereas the frontal nodes of the matching gica #9 are shifted anteriorly and inferiorly to the middle frontal gyrus.

**Group ICA maps and estimated age-related differences in their layout**. The results of the conventional group-ICA factorisation in the in-utero sample are shown in Fig. 4a. The appearance of the spatial maps suggest that they inherit certain signal properties that had previously been revealed in the univariate analysis. Thus,

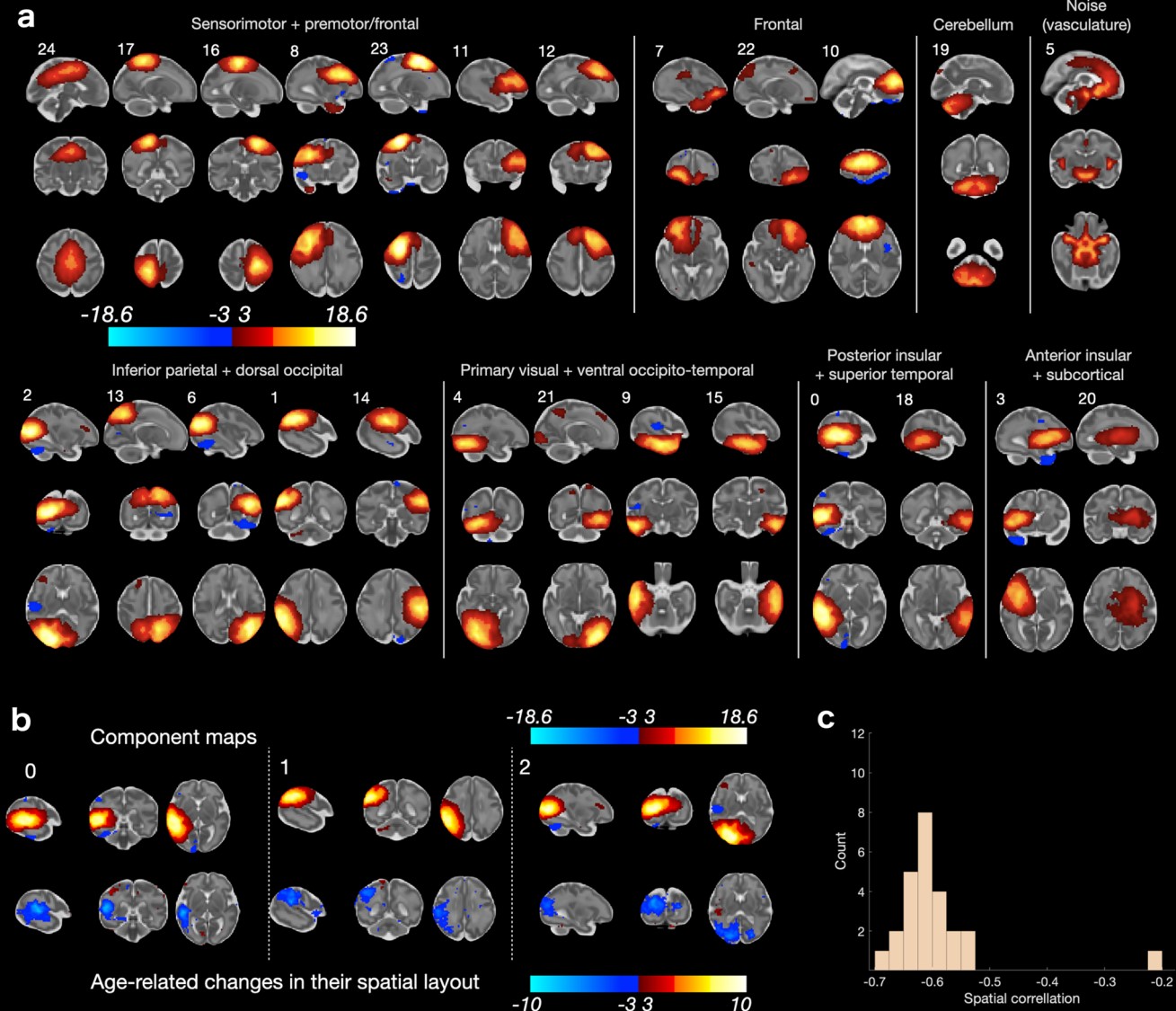

**Fig. 4 Results of group-ICA analysis.** All spatial maps are shown in radiological orientation. **a** Z-scored group-level spatial maps. **b** Spatial maps of the first 3 components (upper row) and corresponding t-maps of age-related changes (lower row), corresponding to the output of the mass-univariate modelling step in Fig. 1a. A negative effect of age can be observed in the most representative component voxels. **c** Distribution of spatial correlations between component spatial maps and corresponding t-maps of age-related changes. The outlier is the component with likely vascular origin (component #5).

their "blurry" appearance is reminiscent of the increased local signal correlations observed in univariate maps of seed-to-brain group-average correlations. In addition, the location of the peaks in many group-ICA maps tended to be biased away from the cortex towards the white matter and a local low-to-high ramp of the component values could often be traced along the boundary between grey and white matter tissues (Supplementary Fig. 6). Despite the above characteristics, most components have anatomically plausible layouts, encompassing a diverse range of functionally relevant areas. The components where peaks were most firmly located within cortical ribbon, were found in sensorimotor and pre-motor areas (e.g., components #16, 17, 23, 24).

Meanwhile, the analysis of age-related changes in the spatial layout of the networks using the dual regression approach (mass-univariate modelling step in Fig. 1a) appear to be affected by a specific bias, as shown using the examples of the spatial maps of the first 3 components and the corresponding maps of the age effect in Fig. 4b, demonstrating a negative effect of age (i.e., a decrease of connectivity with age) in the most representative

component voxels. This somewhat counter-intuitive pattern was observed for all group-ICA components. As Fig. 4c shows, there was a high negative spatial correlation between component group-lCA component spatial maps and corresponding t-maps of the age effect. This pattern appears to be a direct consequence of the signal properties, intrinsic to these data and earlier highlighted in the context of the univariate analyses, showing that there is a negative association between age and strength of correlations for voxels surrounding a seed.

**Maturational networks (matnets).** The above analysis demonstrates an inability to reconstruct coherent maturational relationships in the fetal fMRI data using tools that are widely used in standard network analysis in pediatric and adult populations. In the current and the following sections, we will show that the matnet analysis, built around dense connectomes as an input, is able to overcome this issue and demonstrate comprehensive features of the emerging brain connectivity.

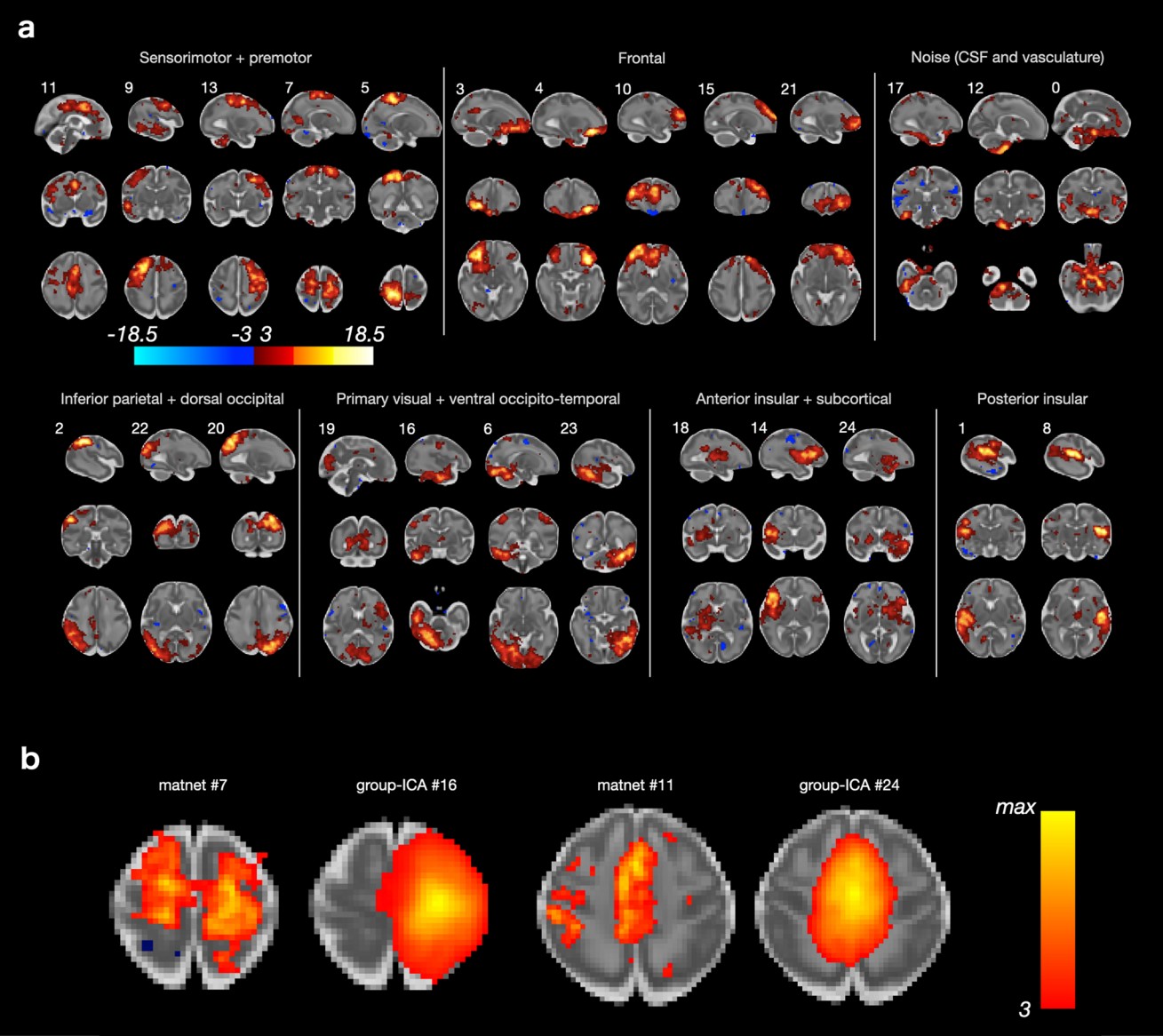

**Fig. 5 Results of maturational network analysis.** All spatial maps are shown in radiological orientation. **a** Z-scored spatial maps, thresholded at abs(z) = 3. **b** Examples of components from maturational and group-ICA analyses, showing that the former tends to show more anatomically specific variation in intensity than the latter. See Supplementary Fig. 12 for all pairs of group-ICA and matnet components.

Thus, results from the maturational network factorization, presented in Fig. 5a, reveal spatial configurations of a high anatomical validity, including locality within the grey matter (Supplementary Fig. 7). In order to ascertain the robustness of the method, we repeated the analysis in approximately age-matched split-half samples, computing matnets in each sample independently, and found a good replicability of the component spatial properties (Supplementary Fig. 8–11).

A qualitative comparison to the paired group-ICA components (for the complete set - Supplementary Fig. 12) demonstrates both the increased spatial specificity of the matnets approach and the differing sensitivity to interhemispheric and distal patterns of network participation. For instance (Fig. 5b), the main node of matnet #11 spatially overlapped with that of group-ICA #24 but in addition encompassed areas in lateral central and pre-motor cortices. Another example is the bilateral matnet component #7, in which the left-hemisphere sub-division overlapped with a spatially compact group-ICA component #16. The more anatomically specific local variations of intensity compared to

the group-ICA maps are reminiscent of the spatial specificity in the age-effect seed-to-brain maps from the univariate analyses. For instance, the matnet map #7 in Fig. 5b has multiple poles, distributed across the somatosensory, motor and premotor cortices, which suggests an early integration of local circuits supporting different functions. In contrast, group-ICA components were typically characterised by a tendency to have only one centre-of-gravity.

**Whole-brain maturational relationships.** Earlier we noted a distinction between (1) matnets proper (i.e., spatially independent maps, obtained by factorisation of the dense maturational connectome) and (2) their emerging connectivity profiles (i.e., age-related changes in connectivity between matnets and all voxels in the brain), which differentiation effectively determines matnets partitioning.

From a biological perspective, matnets delineate areas which have similar targets for their emerging functional connections.

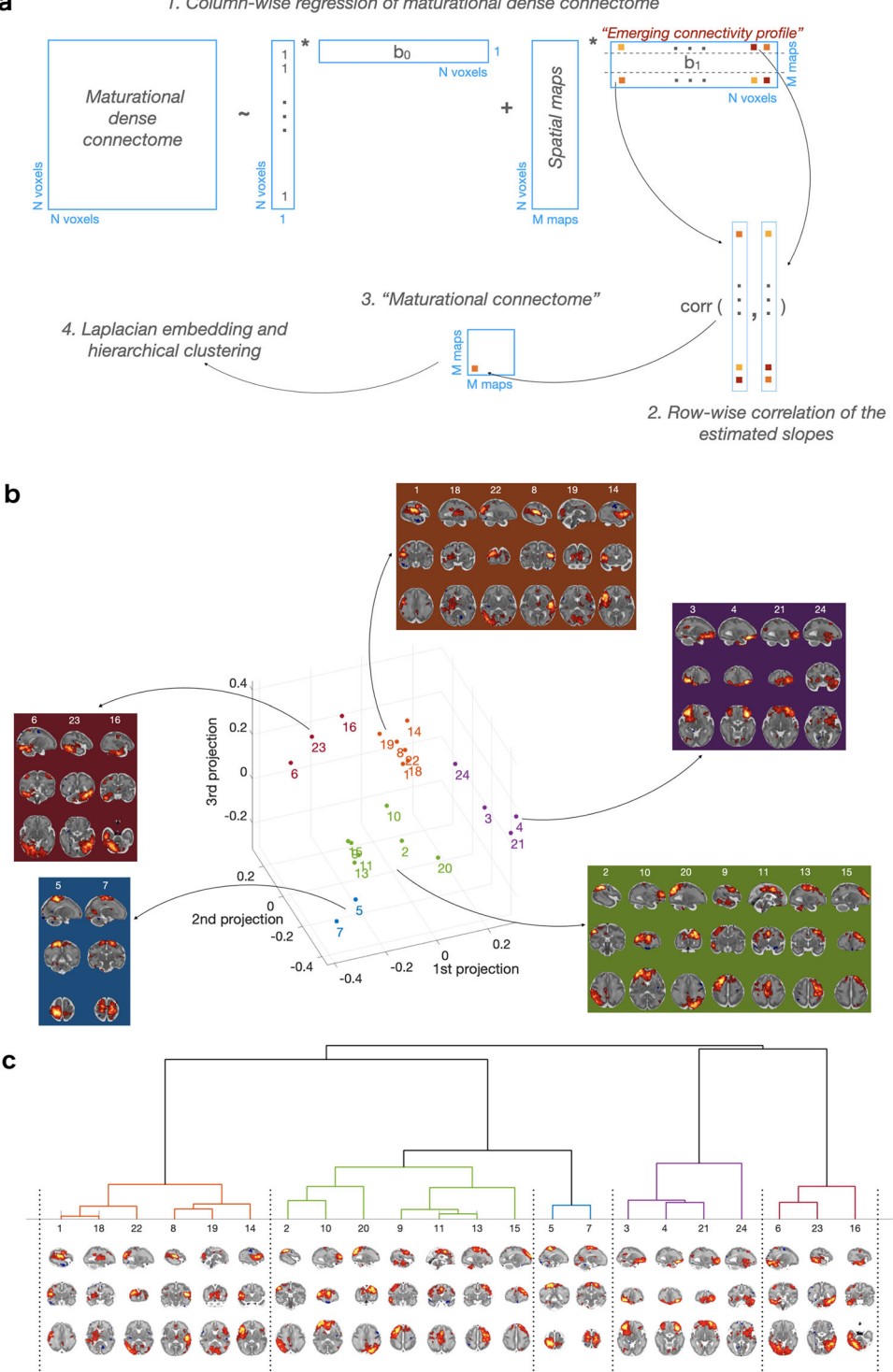

**Fig. 6 Maturational connectome. a** Pipelines for derivation of emerging connectivity profiles associated with matnets and (shown with arrows) the analysis of maturational connectome. **b** Maturational connectome embedding and their split into groups, based on hierarchical clustering. **c** Hierarchical clustering tree.

Alternatively formulated, matnets can be viewed as independent "sources" of emerging connectivity, where their linear mixture determines age-related changes in connectivity of each voxel in the brain. The dichotomy between matnets and their connectivity profiles gives rise to a dual view on the maturation of functional connections which we now consider in detail.

In an analogy to the computation of component temporal courses in the standard-approach using the DR1 step (Fig. 1a),

emerging connectivity profiles associated with matnets are computed as a matrix (here M = 25 components by N = 53,443 voxels) of regression slope coefficients by regressing matnet maps against columns of the thresholded maturational dense connectome (Fig. 6a). This matrix can be treated in two ways.

Firstly, a matrix of pairwise correlations between rows of the connectivity profile matrix summarises a similarity between matnets emerging connectivity profiles, in a similar way as a

matrix of correlations between component timecourses outputted by DR1 (so called "netmats"[30]) characterises functional connectivity between brain networks within the standard group-ICA + DR approach. This provides a whole-brain characterisation of the emerging functional architecture of the in-utero brain, which we call "maturational connectome" for conciseness (Fig. 6a). A three-dimensional embedding of the maturational connectome (Fig. 6b), allows one to appreciate its generic structure. Here a point in space indicates a relative location of a network with respect to other networks, with a shorter distance between networks being indicative of a greater similarity between their emerging connectivity profiles. Several groups of networks, based on the networks' location in the embedded space, can be identified using hierarchical clustering (Fig. 6c). In further analyses we used a 5-group partitioning which was the finest partitioning that did not produce single-network groups. The first group (coded brown) consisted of networks that combined the posterior and anterior peri-insular areas with occipital, auditory and ventral sensorimotor areas. The second group (coded green) consisted of two smaller sub-groups: one comprising dorsolateral pre-motor, dorsolateral prefrontal and medial pre- and supplementary motor areas; the other combining frontal anterior cingulate with inferior parietal and superior lateral occipital cortices, extending into medial posterior areas (precuneus). Adjacent to this group, there was a two-network group (coded blue), comprising dorsal sensorimotor areas. The fourth group (coded violet) comprised ventral frontal and orbitofrontal areas. Finally, the last group (coded purple) combined ventral occipito-temporal areas with dorsal parietal and sensorimotor areas.

Secondly, the alternative view on the matrix of the connectivity profiles associated with matnets is made possible by the fact that one dimension of the estimated regression coefficient matrix is equal to the number of voxels and therefore this matrix represents a collection of complementary spatial maps, that depict targets to which corresponding matnets tend to develop connections to in an age-related manner, or to put it simply, the maps of the targets for their emerging connectivity. From this perspective, clustering matnets into 5 groups is determined by a spatial similarity of their complementary maps. This fact permits an identification of "maturational hubs" for each group as maps that characterise shared connectivity profiles within each group of matnets, for instance, by means of principle component analysis.

Figure 7a summarises these results by showing pairs of matnet-complimentary maps as well as the first principal component maps of complementary maps in each groups. Thus, one can observe a likely vascular contribution in group 1 and 2, evident by the fact that the maps contain areas overlapping with the circle of Willis. In parallel, both maps also contain brain areas which are spatially distinct from the areas reflecting vascular development. For group 1, these areas are bilateral dorsal somatosensory and adjacent parietal cortices and bilateral cerebellum. For the group 2, prefrontal group, the hubs are located in bilateral IFG and superior bank of anterior STG, bilateral insula, bilateral STS. For the sensorimotor (3rd) group, the hubs are not expressed well but some preferential connectivity to right insula and (predominantly right) striatum and thalamus can be observed. For the 4th (ventral frontal and orbitofrontal) group, the hubs were located in the bilateral SFG and MFG. Finally, for group 5 (ventral visual stream areas), the hubs were located in (predominantly right) lateral parietal and dorsal parieto-occipital cortices and right posterior perisylvian cortices.

Furthermore, as a proof-of-principle that maturational relationships are determined by an age-related *increase* of connectivity and as well as a demonstration of the potential application of the method to the study individual variability and inter-regional trajectories, the following result can be

presented. Here, we estimated the temporal coupling between matnets and their complementary maps as a function of age. For this, both the matnets and their complementary maps were thresholded at z > 5 in order to reduce a degree of potential spatial overlap between the two and their time courses were computed as weighted averages of the above-threshold voxels. Figure 7b shows the results for two maturational groups, with differing age-related trajectories, whereas age-related trajectories for all 25 components are shown in Supplementary Fig. 13.

## Discussion

In this paper, we presented an analytical framework that characterises functional networks as an emerging property of the brain. Within this framework, the fusion of voxels into a network is determined by the similarity of their maturational profiles with respect to the rest of the brain. In effect, this represents a computational implementation of Flechsig's principle[21] that states that concordant maturation characterises functionally related areas. In an implicit form, Flechsig's principle has been previously utilised in the studies of structural covariance[31] in developmental cohorts[32,33], including fetuses[34]. Here we apply the principle explicitly to the study of emerging functional organisation in the in-utero brain.

We also tested the performance of the framework in the neonatal dHCP sample. Overall, matnets showed excellent agreement with group-ICA analysis of the same data. Furthermore, matnets revealed features characteristic of more mature brains with a greater specificity, such as more symmetrically distributed patterns across the two hemispheres and a nearly complete default mode network. Conceptually, a greater fractionation of group-ICA neonatal networks is not surprising, because compared to the "connectivity-as-present" representation ICA provides, matnets reconstruct maps of "connectivity-in-making".

Further fractionation of the networks into separate areas was observed in the analysis of the fetal brain connectivity. Here the difference between a group-ICA and matnet approaches becomes even more prominent. We have showed that maturational networks (Fig. 5) permit identification of spatially distributed patterns of connections with a remarkable anatomical specificity for the in-utero data, owing to their reliance on the benign signal properties that reveal an age-dependent increase of mid- and long-distance connectivity in a spatially selective manner. We have also showed that maturational networks represent a coherent way of characterising maturational patterns in the context of fetal fMRI, compared to inference using the standard approach (Fig. 4), in which results appear to be affected by specific biases (we will discuss this below).

Compared to ex-utero data, in-utero fMRI data inherently suffers from decreased signal-to-noise ratio and greater artefacts which contribute to difficulties identifying distributed networks in the fetal brain. Nevertheless, the matnet results indicate that fundamental features of neonatal and even adult-like functional architecture occur prior to the exposure to extrauterine environmental influences. This is reflected in a range of motifs characteristic of the neonatal brain connectivity, which can be viewed as the eventual target for maturational processes in utero. Thus, several networks revealed a non-negligible bilateral component, that agrees with the studies of pre-term and term born babies[4–6], as well as in-utero seed-based connectivity fMRI studies[8], suggesting that interhemispheric coupling becomes established during this period. The maturational networks also characterised a range of non-trivial functional relationships that are similarly observed in neonatal data[6], such as functional associations between the inferior parietal regions and precuneus; between the anterior cingulate cortex and lateral orbito-frontal cortex,

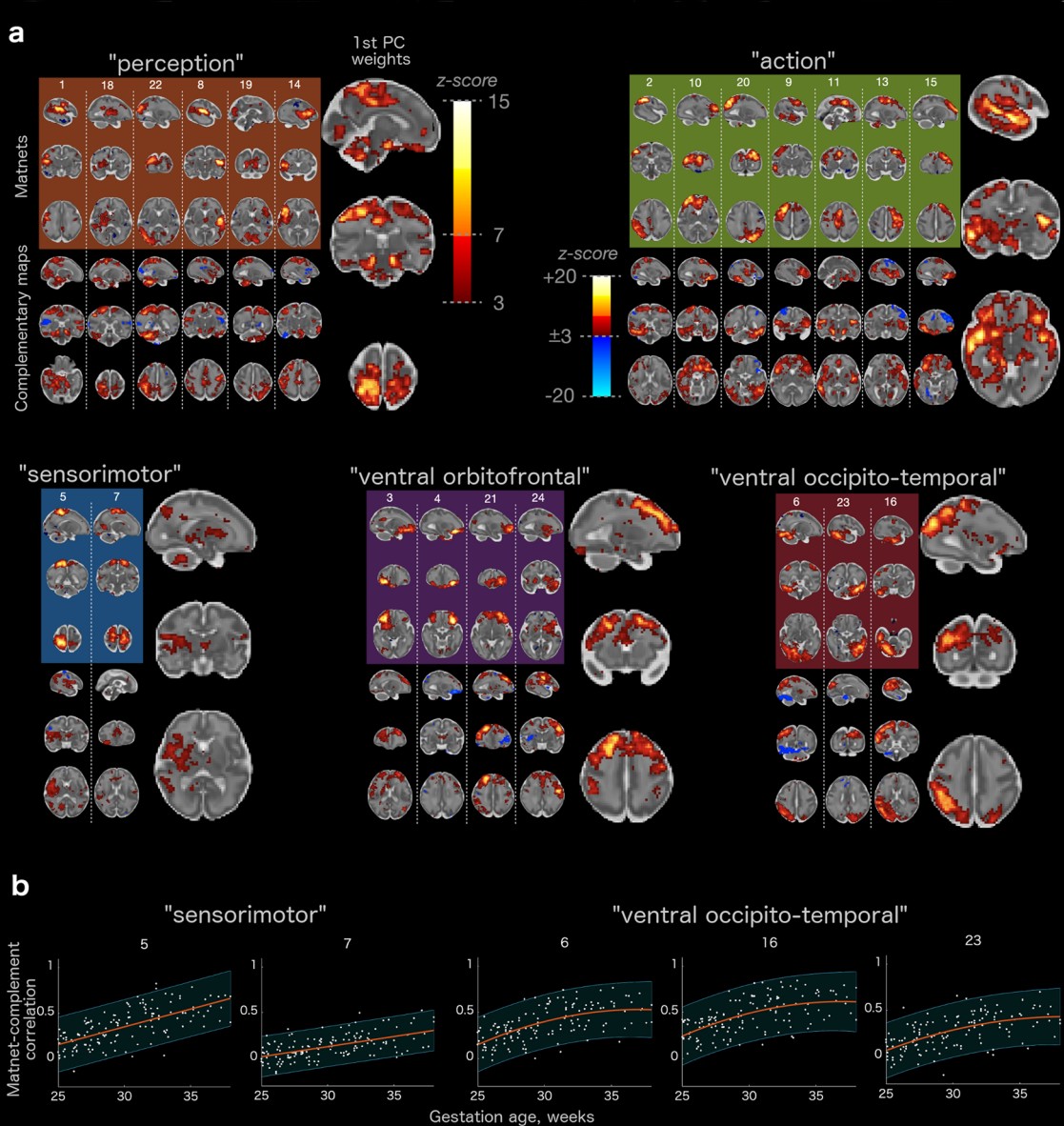

**Fig. 7 Matnets and their complementary maps. a** Spatial maps of matnets (top), their complementary maps (bottom), and the 1st PC of the complementary maps (right) in each matnet group. **b** Examples of temporal correlations between time courses of matnets and their complementary map for two matnet groups (see Supplementary Fig. 13 for all maturational components). Lines represent the best-fitting polynomial models and shaded regions are confidence intervals (alpha = 0.05).

between the medial and lateral (pre)-motor cortices; between the central sulcus and posterior insular cortex; or between the dorsal and ventral stream regions. This demonstrates that these emerging functional relationships across spatially distinct regions are an intrinsic property of the brain and provides crucial validation of the findings of neonatal studies where the complementary role of environmental influences had been unclear.

An additional level of insight into the developmental sequelae of the fetal functional brain and the shaping of future network architecture is provided by considering matnets in association with their complementary maps, with the latter characterising the matnets' emerging connectivity profiles. This leads towards two novel constructs: the maturational connectome, that summarises similarity of emerging connectivity profiles between pairs of matnets (Fig. 6), and maturational hubs, that represent common targets for the matnets's maturing connections (Fig. 7). Together, their analyses allow us to characterise macroscopic patterns of

connectivity that emerge during this critical stage of human development.

A conspicuous generic feature of the maturational connectome, revealed by its low-dimensional embedding, is the tendency for homologous contralateral networks to cluster together. Overall, the clustering analysis identifies two larger groups that occupy the central location in the embedded space and three smaller, more peripheral, groups. Based on the areas that dominate their anatomical layout, the three smaller clusters of networks can be labelled as orbitofrontal, ventral visual and sensorimotor groups. Of the larger groups, one was dominated by the cortical nodes of perception and bodily sensation (occipital, auditory and somatosensory limbic areas) but also included nodes in the motor and motor limbic[35] (anterior cingulate and anterior insular) cortices. The other larger group was dominated by the functional nodes responsible for an environmental interaction through action (dorsolateral and medial pre-motor cortex and pre-frontal areas),

but also included a sub-group of networks which spatially overlap with nodes of the future default mode networks[27,36], such as precuneus, anterior cingulate and angular gyrus. Notably, the location of the latter within the embedding space was midway between the notional perception group and the remaining networks of the notional action group, hinting both towards hub connectivity patterns and their apparent role in modulating internal and external inputs whilst mind-wandering or performing cognitively demanding tasks later in life[37].

The framework also allowed us to describe maturational hubs of the in-utero connectome, characterised as regions in which networks within a matnet group form preferential connectivity in an age-related manner. Areas in the dorsal somatosensory and adjacent parietal cortex which process sensation and spatial information, as well as the cerebellum, were identified as hubs for the first group of matnets, combining perception, bodily sensation and motor limbic areas located outside sensorimotor cortices, suggesting integration of information across different perceptual and limbic domains towards a central cortical processing unit. The analysis also reveals an important role of high-level associative areas within the brain connectome from the onset of the brain functional development. Thus, the hub for ventral occipital and temporal areas, which in the adult brain encodes representations of abstract visual information, is found in the posterior parietal cortex, including the right IPS and the posterior node of ventral attentional network (VAN)[38]. This supports evidence, similarly observed at the level of individual matnets (e.g., matnet #6), of an ongoing integration of the ventral and dorsal stream representations. We also observed an emergence of links between ventral action-related limbic areas, representing internal motives and drives in the adult brain[39], and areas associated with encoding representations of abstract rules for goal-directed behavior and with executive control. This is made evident by the fact that the major hubs for the action group, which among others included networks in lateral prefrontal cortex, were located in the anterior perisylvian and insular cortices, posterior ventral orbitofrontal and anterior temporal cortices, overlapping with limbic and motor limbic cortices and the prefrontal hub of the adult VAN. Reciprocally, the dorsal prefrontal areas implemented in the adult dorsal attentional network (including frontal eye field), working memory and executive control were identified as a hub for ventral orbitofrontal matnet group, which in the adult brain are known to project feedback pathways to the dorsolateral prefrontal cortex, providing the latter with information on internal environment[40]. These findings indicate that the neural machinery for linking decisions and actions to internal wishes and motives start emerging as early as the fetal period in human life.

These results challenge the view that the transition from fetal to a more mature functional architecture is manifested by the shift of functional hubs from primary to associative areas[41,42] and aligns with earlier studies of structural connectivity in full-term and preterm neonates showing that adult-like features of the structural connectome can already be observed at this early period[43]. They also align with a recent study showing an early patterning of deep projection neurons in the frontal lobe, which could provide a structural infrastructure for the functional connectivity of high-level associative areas[44]. The distinction between more mature and in-utero functional connectome features appear to be signified by a relative disconnection along posterior-anterior axis, such as between nodes of fronto-parietal networks which are considered responsible for the integration of information across behavioral domains in the adult brain. Similarly, consistent with previous findings in preterm neonates[4], no strong evidence of the links between medial posterior and anterior nodes of the default mode network were observed, which at this period appear to be integrated, respectively, within parietal and frontal networks and

lack a prominent role in the observed whole brain functional architecture. Interestingly, a nearly complete default mode network identified with matnets in neonates contained only a small cluster of voxels in the medial parietal cortex, supporting the evidence that a fully functional DMN may emerge only as late as at the age of 3[45].

In-utero maturation is associated with competing physiological processes which may potentially leave a footprint on the properties of the fMRI signal[46,47], thereby raising a question about the biological underpinnings of the age-related signal changes implicated in the derivation of maturational networks. For instance, one cannot exclude the possibility that changes in the long-distance connectivity, in the absence of a mature structural connectome, are in part due to the coordinated development of the brain's vasculature[48]. De-confounding the latter from the estimates of neural connectivity is a contentious issue even in the context of adult resting-state imaging[49,50]. In the fetal brain, the problem may be further exacerbated as the development of brain neural systems goes hand in hand with the development of other organ functions including the vasculature and thus are likely collinear to the degree that the two are indistinguishable at a level visible to fMRI.

The effect of tissue composition on the T2* relaxation rate may also represent an intrinsic confound for our analysis. The dHCP acquisition utilises a substantially longer TE (60 ms) compared to a benchmark adult acquisition (e.g., HCP protocol: 33 ms[51] in order to align with longer relaxation rates in the developing brain. However, assuming $T2^* = 100$ ms for neonates[52] and that matching TE and T2* may (theoretically) provide higher SNR, the current TE may be a more "optimal" choice for the older fetuses, therefore, potentially biasing estimates of the age-effects. However, this is not supported by the observed tendency of white matter seeds/voxels to show a negative association with age compared to the cortical regions, given that age-related tissue changes are likely to be more pronounced in the white matter than in the cortex, as white matter maturation occurs throughout gestation and myelination does not commence in many regions until the early neonatal period[53]. One would then expect greater positive age-related changes for the white matter if the effect was due to the SNR-TE relationship.

Another potential confound is that fetuses tend to change position from pointing upwards to head-down position later in the gestation, potentially affecting the signal. However, this factor cannot explain age-related increases in connectivity leveraged by the matnet analysis, as the head-down position would result in a decrease in SNR and consequently decreased estimates of connectivity strength, due to the effects of the surroundings such as the adjacent bones and air-filled bowels.

Finally, the registration accuracy represents a fundamental issue, that can never be completely resolved by nature of the changing fetal brain. To ameliorate this issue, we used a very comprehensive approach to the group-space registration, previously exploited for the neonatal data[6], which avoids a necessity of computing large – and potentially error-prone – deformations and at the same time achieving a remarkable alignment even for morphologically distant brains (see Methods for the description). In general, we expect that inaccuracy in registration will to some degree be balanced out between ages by diverging factors: in younger a cause of misregistration is likely to be a simple brain morphology that lacks distinct landmarks; in older fetuses it is the unique complexity of gyrified brain that makes it difficult to fit a standard space. However, further work is needed to fully assess the effects of the template choices and registration procedures for this challenging type of data.

Compared to maturational networks, group-ICA components identified with a standard group-ICA approach had diminished

spatial complexity and anatomical specificity and were biased towards the white matter. Notably, the results of dual regression modeling showed that local connectivity within group-ICA networks diminishes with age. Such characteristics fit well those of the functional nodes described in the fetal animal studies, which center on the cortical subplate and act as local amplifiers of the thalamic activity with spread that does not conform to anatomical boundaries[3,54]. This may suggest that group-ICA and maturational networks truthfully reflect two different states of the fetal functional brain: a truly "fetal" subplate-centered[55] and locally active state depicted by the group-ICA, that gives way to the adult-like cortex-centered and spatially distributed state of maturational networks. Against this intriguing interpretation, though not necessary incompatible with it, are the results of the univariate analysis of the connectivity metrics. The latter demonstrates that the correlational structure of the data, that underlies the derivation of the group-ICA components, is dominated by a spatially smooth and non-linear distance-dependent gradient, which scales negatively with age. The factors that make biologically-motivated interpretation of this gradient unlikely is the spatially indiscriminate character of these phenomena combined with a violation of anatomical boundaries, including the large connectivity distance between the two brain hemispheres which in reality are separated by a CSF filled inter-hemispheric fissure.

An initial hypothesis to explain the origins of distance-dependent gradients and its interaction with age can be based on the potential contribution of two factors: motion and effective resolution. The role of motion on connectivity estimates has been demonstrated in adult imaging, where it has been shown to decrease long-range connectivity and overestimate local connectivity[19,20]. Although we used a comprehensive image processing pipeline to account for head motion during data acquisition, fetal imaging data is still especially susceptible to this effect as fetuses have virtually no motion-free periods. Even if the fetus stays still, maternal breathing cycles and endogenous motion in the non-rigid tissues surrounding the fetal head continue to cause a constant change of position. Under these circumstances, effective resolution naturally leads to age-related differences in the effect, which likely explains the dual regression result showing a decrease in connectivity with age within the most representative component voxels. The brain undergoes a 3-fold growth in size over the studied period, which implies that real-world separation between pairs of voxels in a standard space is smaller for younger subjects than for older ones and thus a greater effect of distance as measured in the common space. In light of the differences in signal properties between the grey and white matter and their modulation by age, the possible contribution of other factors such as modulation of the BOLD signal itself and/or the role of age-related changes in tissue content also should not be disregarded.

Below we outline several limitations of the study. First, the current study has the well-known limitations of cross-sectional analyses whereby between-subject variability can be confounded with aging effects. Nevertheless, cross-sectional data are expected to dominate fetal research for a foreseeable future, as scanning mothers during pregnancy on multiple occasions presents both ethical and practical challenges. In the meantime, one can strive for better estimates of cross-sectional trajectories, using improved modelling and larger data samples. Our results are based on one of the largest fetal fMRI data sets both in terms of the number of subjects and the number of volumes per subject. However, further improvements in modelling can be achieved when data for the full fetal dHCP cohort will be made openly available to the neuroscientific community in the coming year. This would increase the current data sample by a factor of nearly 2.

The second limitation concerns generalization of our conclusions to other data samples, especially in the context of fetal fMRI as a novel field, where norms of data acquisition are yet to be established. Unfortunately, fetal fMRI has not as yet stepped in into the age of normative open-access big data[56] which has enabled recent progress in the study of ex-utero connectivity, (e.g.[51],). However, the qualitative comparison of our results with the results drawn from other studies gives us a certain confidence that our results are not specific to our sample. For instance, there was a remarkable similarity between our group-ICA results and the group-ICA results reported in a recent paper[13], despite considerable differences in the acquisition sequence (multi- vs single-band), spatial image corrections (dynamic distortion and slice-to-volume corrections vs volumetric alignment only) and de-noising pipelines (predominantly motion parameter-based vs. ICA-based). Furthermore, the qualitative characteristics of group-ICA components as well as the dominance of distance-dependent gradient over the correlational structure also appear to be reproducible across the studies[13].

In conclusion, we describe a novel framework that delineates the emergence of resting state networks in the fetal human brain with remarkable spatial specificity and provides a comprehensive model of inter-areal maturational relationships, assigning a central role to the brain regions associated with active environmental interaction through perceptual and motor-planning mechanisms. A discerning feature of this maturational network framework is a prospective incorporation of the variable-of-interest (here, age) into network estimation. This can potentially make the method adaptable to other applications, such as studying early human development through childhood, network maturation in neurodevelopmental disorders such as autism, ageing and exploring the connectivity underpinnings of changing patterns of behavior across the lifespan.

## Methods

**Data**. Participants were prospectively recruited as part of the developing Human Connectome Project, a cross-sectional Open Science initiative approved by the UK National Research Ethics Authority (14/LO/1169). Written informed consent was obtained from all participating families prior to imaging. At the time of the study initiation, resting-state fMRI data were acquired in 151 fetuses older than 25 weeks of gestation (62 females, 77 males, 5 unknown), median age = 29.5w, range = [25 38], with Philips Achieva 3 T system (Best, NL) and a 32-channel cardiac coil using a single-shot EPI (TR/TE = 2200/60) sequence consisting of 350 volumes of 48 slices each, slice grid 144 × 144, isotropic resolution = 2.2 mm, multi-band (MB) factor = 3 and SENSE factor = 1.4[29]. All fetal brain images were reported by a neuroradiologist as showing appropriate appearances for their gestational age with no acquired lesions or congenital malformations of clinical significance. Data from 7 fetuses did not pass visual quality assessment due to excessive motion and failure in image reconstruction.

The data of the remaining 144 fetuses were preprocessed using a dedicated pipeline[24–26]. In brief, the data underwent MB-SENSE image reconstruction, dynamic shot-by-shot B0 field correction by phase unwrapping and slice-to-volume (S2V) motion correction[24]. The data were then temporally denoised using several sets of confound regressors, aiming to address various types of artefacts. The denoising model combined volume censoring regressors, aiming to reject volumes (at a heuristically selected threshold) (Supplementary Fig. 14), highpass (1/150 Hz) filtering regressors of direct cosine transform matrix in order to remove slow frequency drift in the data, 6 white matter and cerebrospinal fluid component timecourses (obtained using subject-level ICA within a combined white matter + CSF mask, (e.g.[57]), and 3 variants of voxelwise 4d denoising maps in order to account for the local artefacts in the data: (1) folding maps ($N = 2$) which aggregate time courses of voxels linked in multiband acquisition to voxels in the original data, aiming at filtering out leakage artefacts; (2) density maps, representing temporal evolution of an operator that compensates for the volume alterations a result of distortion in phase encoding direction, and aiming to filter out residual effects of distortion correction on the voxel timecourses; and (3) motion-parameter-based regressors, expanded to include first and second order volume-to-volume and slice-to-slice differentials as well as their square terms, aiming to remove motion-related artefacts[58,59].

**Neonatal sample and data**. The characteristics of the scanning sequence for the neonatal data, which were acquired using the same hardware as the fetal data, are described elsewhere[5,6]. The data were preprocessed using dHCP neonatal pipeline[6].

For the current analyses we created a sample which ages were symmetrically distributed around 37.5 gestation weeks, i.e., approximately the age of the oldest subjects in the fetal sample (mean age 37.27, sd = 3.98). The complete dHCP cohort is not symmetrical (Supplementary Fig. 5) and heavily skewed to the older ages. To compensate for this, we included all participants that were younger than 37 gestation week old and then randomly sampled participants of older ages to create a near-symmetrical distribution. 311 participants were selected for the analysis.

**Registration to the group space**. A 4D atlas of the developing brain (available at https://brain-development.org/brain- 594 atlases/fetal-brain-atlases/)[60] was used as a template space for data registration. A schematic depiction of the registration to a common template space is shown in Supplementary Fig. 15a. The mapping between a functional native space and the common template space is constructed as the concatenation of several intermediate transformations, which ascertain a gradual alignment between spaces to minimise risks of gross misalignment as a result of the substantial differences in the brain topology across the range of gestation ages (Supplementary Fig. 15b)[6]: (1) rigid alignment between mean functional and anatomical scans calculated using FLIRT boundary-based registration[61]; (2) a non-linear transformation between an anatomical T2 scan and an age-matched template calculated using dual-channel (T2w and cortex) ANTs[62]; (3) a sequence of non-linear transformations between templates of adjacent ages (e.g., 24 and 25, 25 and 26, etc.), also calculated by ANTs. These transformations were concatenated to create a one-step mapping between functional and group template space, that allows us to project between native and template spaces with a single interpolation. The template corresponding to GA = 37 weeks was selected as a common space for group analysis based on the considerations that it has a greatest effective resolution and topological complexity. An additional group space was created by symmetrizing the GA = 37 week template with respect to the brain midline, with appropriate adjustment of the mapping from native spaces, that included an additional non-linear transform from non-symmetrical-to-symmetrical template spaces. After registering the functional MRI data to the template space, they were smoothed using 3 mm Gaussian kernel. No lowpass filtering was applied in the temporal dimension.

**Univariate data analyses**. For the illustrative analyses, presented in Fig. 2a and Supplementary Fig. 3 & 4, the seeds for the seed-to-brain analysis were determined empirically using the results of modelling age-related changes in interhemispheric connectivity between pairs of homologous voxels (Supplementary Fig. 16), performed in the symmetrical template space[63]. The subject-specific maps of homologous voxel connectivity were obtained by calculating the correlation between timecourses of homologous voxels in the two hemispheres. The age-effect map was obtained via a voxel-wise regression with age as a covariate. The seeds for grey matter were created by thresholding the age-effect map from the above analysis at z > 3, which rendered 3 sizable clusters of voxels (14, 32, and 45 voxels). Given the absence of positive age-related increase in connectivity between homologous voxels for white matter areas, the white matter seeds were created by thresholding the age-effect map of interhemispheric connectivity negatively at z < −3, and then manually adjusting clusters to fit the size of the grey matter clusters. Because the seeds were defined in the symmetrical template space, the seed-to-brain connectivity analysis was also performed in this space. The seed-to-brain group-average correlation map was calculated by first calculating individual maps of correlations between time course of a seed and time courses of all voxels in the brain and then averaging these maps across subjects. The age-effect map was obtained by fitting individual maps voxelwise using age as a covariate.

For the analysis of the relationship between similarity of seed-to-brain maps and the distance between them, cortical mask was parcellated into 300 clusters with k-means algorithm using voxel coordinates as input. The seed-to-brain group-average correlation and age-effect maps were calculated as above. Spatial distance between a pair of parcels was computed as a distance between their centres-of-gravity.

**Group-ICA**. The derivation of group-average modes-of-variation and their subject-specific variants was performed using the protocol of FSL MELODIC for group-ICA analyses[17], including FSL MELODIC's Incremental Group Principal component analysis (MIGP step)[14], and the standard procedure of dual regression, implemented in FSL[64]. The number of derived components was set to 25, in accordance with the published research in neonates[6].

**Maturational modes of variation**. The pipeline for derivation of maturational modes of variation is shown in Fig. 1b. First, a symmetrical matrix of correlations between each pair of voxels in the brain mask was calculated, aka "dense connectome", for each subject separately. Each element of the dense connectome was fitted across subjects with age as covariate, rendering a voxel-by-voxel matrix of age-effect beta coefficients. The matrix was then converted into t-values, rendering maturational dense connectome, subsequently thresholded at 0 in order to leverage the age-dependent increases in correlations in network estimation. The rationale for positive thresholding is described in the Results section. In order to perform connectome factorisation, an intermediate step of dimensionality reduction, analogous to the MIGP[14] step of the group-ICA, was applied. For this, the

maturational dense connectome (size: N voxels by N voxels) was split column-wise into 200 blocks (size: N voxels by N voxels/200. At the initial step, a matrix consisting of the first two blocks was formed and subsequently reduced to 500 components using singular value decomposition. An iterative procedure was then run that consisted of concatenating the current matrix of 500 components with a following block and subsequent reduction to 500 components by SVD, until all blocks were exhausted. The output of this procedure was used to obtain the final factorisation of 25 components using FSL MELODIC.

We also considered whether a measure of a global motion (framewise displacement (FD)) needs to be included as a covariate, given that the motion of older fetuses may be constrained by their own size and upside-down position. For this, a global measure of frame-wise displacement (FD) was calculated in the following steps. First, a mean of absolute FD for each motion parameter was calculated across time (altogether 96 values: 6 rotations + translations times 16 multiband stacks) in each subject. These means were collected into a 144 (number of subjects) x 96 matrix, which was then z-scored across rows (subjects). Finally, the first principal component was computed and used as a measure of global between-subject variation in motion.

We found that a small-effect correlation between age and FD, $r = −0.25$. Consequently, we analysed whether a potential confounding effect of FD alters age-effect statistics in a spatially varying manner, to which, unlike to a global effect, the ICA factorisation would be sensitive. An alternative hypothesis is that FD is not an independent factor but alters age-related statistics only because it is collinear with age. For this we considered age-related changes in interhemispheric connectivity between homologous left and right voxels.

First, we found that the maps of age-effect statistics computed with and without global FD as a covariate are highly correlated, $r = 0.98$. Furthermore, the inclusion of FD as a covariate resulted in a graded decrease of estimates of age-effect statistics with respect to the magnitude of the estimated age-effect (spatial correlation between age-effect t-map calculated without FD as a covariate and the difference between maps calculated with and without FD as a covariate: $r = −0.49$). In other words, the FD inclusion makes negatively values less negative and vice versa for positively values. Finally, we note the age effects tend to be tissue specific, i.e., tended to be more positive in the cortex and more negative in the white matter (Supplementary Fig. 16), which is not expected if the source of association was motion. Taking together, the above observations can be explained based on the hypothesis of FD-age collinearity, whereas an alternative interpretation presuming an independent effect of FD entails a complex interaction between tissues, age and motion, for which we do not have substantial evidence. These considerations serve as a justification for not inclusion of FD in the downstream modelling.

**Maturational connectome analysis**. The pipeline for derivation of the maturational connectome is shown in Fig. 5a. It consists of the regression of the maturational networks against the maturational dense connectome in order to obtain #networks by #voxels matrix of regression coefficients. Correlations between each pair of rows of the matrix were then estimated, collected into a matrix which constitutes the maturational connectome. In order to reveal a structure of the whole-brain maturational relationships, the maturational connectome matrix was embedded into 3-dimensional space as an eigendecomposition of a graph normalised Laplacian. A point in the embedding space indicates a relative location of a network with respect to other networks (i.e., a shorter distance means closer maturational ties). A partition of networks into groups of networks was performed using the Ward method of hierarchical clustering[65], based on the network coordinates in the embedding 3D space.

**Statistics and reproducibility**. In order to ascertain the robust performance of matnets factorisation, the analysis was performed in the neonatal sample, comparing the results to the results of group-ICA. In fetuses, we ran additional analyses in approximately age-matched (mean age: 30.50 (3.23) and 30.42 (3.50), t (142) = 0.14, $p = 0.89$, two-tailed) split-half samples.

**Reporting summary**. Further information on research design is available in the Nature Portfolio Reporting Summary linked to this article.

## Data availability
The minimum dataset that contain input files necessary to reproduce the results of group-level analyses reported in the manuscript are available at https://gin.g-node.org/slavakarolis/matnet_paper. Source and preprocessed individual data, with recent improvements implemented during ongoing pipeline development, is/will be made available in the forthcoming release of the dHCP fetal cohort data (anticipated date of release is June 2023).

## Code availability
The code pertaining to the derivation of the matnets is available at https://gin.g-node.org/slavakarolis/matnet_paper.

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

## Author contributions

Conceptualisation, writing—original draft—V.R.K., J.O.M., E.D., T.A. Methodology, validation—V.R.K., L.C.G., A.F., E.H., A.P., E.D. Visualisation—V.R.K. Formal analysis—V.R.K., S.F., L.C.G., A.F., E.H., A.P. Data curation—V.K., M.P. Investigation—S.R.F., E.H., A.P., M.R. Funding acquisition—D.R., A.D.E., J.H., T.A. All authors contributed to writing – review & editing.

## Funding

The Developing Human Connectome Project was funded by the European Research Council under the European Union Seventh Framework Programme (FP/20072013)/ERC Grant Agreement no. 319456. The Wellcome centre for Integrative Neuroimaging is supported by core funding from the Wellcome Trust [203139/Z/16/Z]. The authors also acknowledge support in part from the Wellcome Engineering and Physical Sciences Research Council (EPSRC) Centre for Medical Engineering at Kings College London [WT 203148/Z/16/Z], the Medical Research Council (MRC) Centre for Neurodevelopmental Disorders [MR/N026063/1], and the Department of Health through an NIHR Comprehensive Biomedical Research Centre Award (to Guy's and St. Thomas' National Health Service (NHS) Foundation Trust in partnership with King's College London and King's College Hospital NHS Foundation Trust). V.K. and T.A. were supported by a MRC Clinician Scientist Fellowship [MR/P008712/1] and MRC translation support award [MR/V036874/1]. J.O.M. is supported by a Sir Henry Dale Fellowship jointly funded by the Wellcome Trust and the Royal Society [206675/Z/17/Z]. L.C.G. is supported by the Comunidad de Madrid-Spain Support for R&D Projects [BGP18/00178]. S.R.F.'s research is supported by the Royal Academy of Engineering under the Research Fellowship programme [RF2122-21-310] and Wellcome Trust [215573/Z/19/Z]. A.E.F. acknowledges additional funding from the UKRI CDT in Artificial Intelligence for Healthcare in his role as a Senior Teaching Fellow [EP/S023283/1].

## Competing interests

The authors declare no competing interests.
