## [Peer Review File · Communications Biology]

Reviewers' comments:

Reviewer #2 (Remarks to the Author):

The work by Karolis and colleagues deals with the fundamental challenge of characterizing functional brain development before birth in humans. Despite some progress in this field, this remains an elusive research goal. While fetal fMRI has been the topic of a handful of studies before, this work may stand out for two reasons. First, they relied on a robust and unique dataset of 144 normally developing cases, acquired with a relatively long fMRI (compared to other fetal fMRI papers) sequence. Second, they did not spare the time to develop analytical methods that may efficiently deal with the problem of how to characterize resting state networks that 'appear from nothing' during development. While the focus is not always entirely clear (that is: biology or new methodology), a substantial part of the work describes the analytical framework to study and concludes that it is superior to the standard approaches. The analytical framework is convincing and new.

In the formulation of the analytical framework to study emergence of brain networks, the authors proposed two possibilities. First, by using dual regression – a technique that is well established in the resting-state fMRI community – in which gestational age is a covariate of interest in a multivariate regression model. Second, simply said - a new model where individual elements in the matrix are used in a multivariate regression model with age as covariate, and then combining these into a maturational network. The manuscript is well written, the findings of this work are supported by their data and methodology.

The article could be improved if some of the biological underpinnings were better articulated. While the authors used numerous strategies to overcome fetal motion, it could be further disputed to what extent do these age-related findings originate from the many physiological changes during gestation and not brain functional maturation per se. More details about the last concern are below. The below list are my suggestions for improvements, which are mainly questions to be answered here or in the manuscript. Question number 1 might require more experiments to be done.

1. The authors introduce a new methodology and very convincingly deliver evidence that it is superior to an alternative approach (dual regression). However, they base their conclusion on a single dataset of fetal MRI, which is known to be extremely challenging, and many age-related phenomena, such as vasculature maturation or maternal physiology / movement, are very difficult to control for. The impact of the method on the field could be increased if the authors already utilized it on other data they might be in possession of, for example, on postnatal newborn, preterm or infant datasets. Could the authors please consider this suggestion or make an argument why the same purpose (emerging brain networks) cannot be tested on that dataset, too? Since there is a lot more published in postnatal field, even using EEG/MEG or other modalities, there are higher chances to validate the findings on dynamically changing network structures.

2. Fetal head motion patterns are also changing during gestation, for example by the decreasing space available for the fetus to move. I understand that they used the motion regressors during the signal processing, but was overall/total FD used in the developmental model as an additional covariate? This would perhaps help to counteract the GA-related changes in the movement patterns, which may mimic GA related changes in the functional network architecture.

3. On a similar note, other factors are changing as a function of gestational age. T2 time of the fetal brain is strongly correlated with GA (Counsell et al. AJNR Am J Neuroradiol. 2003 Sep; 24(8): 1654–1660), so for fetal fMRI studies, it might be worth using a variable TE in the range between 130-190 ms. (Blazejewska et al. Magn Reson Med. 2017 Sep;78(3):909-916.) The authors used 60, can they please comment on this? How would this affect the maturational networks?

4. Third, did the authors take into consideration the head position of the fetus? The positioning of the

fetus towards the pelvic outlet towards the end of gestation might decrease the SNR, since the whole FOV is surrounded by more pelvic bones and more air filled bowels.

5. There have been prior works in fetal fMRI using ICA, one of the two methods the authors evaluate in the beginning, so the authors might want to consider comparing their results also with the networks reported in the (not yet cited) publications (e.g. Schopf et al. Int J Dev Neurosci. 2012 Feb;30(1):11-7., Schopf et al. Front Hum Neurosci . 2014 Oct 2;8:775.), or dispute why their methodology may or may not reveal different networks than these publications.

6. The authors write "Therefore indicating that functional connections supporting high-level human cognition start emerging even prior to exposure to the extra utero environment.". It is not yet clear what the author's hypotheses are for the biological underpinning of how higher level human cognition can be present, e.g. multisensory integration? They relate these findings mainly to past papers based on fMRI. However, can they elucidate what mechanism might be driving the establishment of such networks at the neuronal level? There are also other works that proposed that neural correlates of higher cognitive functions may emerge before birth, the authors might want use the findings of these previous works to support some of their hypotheses (e.g. Schopf et al. Front Hum Neurosci . 2014 Oct 2;8:775., Jakab Front Hum Neurosci. 2014 Oct 22;8:852.). Can the authors maybe explain these age related changes by some data on structural connectivity, e.g. to show when long-range association fibers emerge? How would regional changes in the resolution of the subplate and subplate-cortex connections affect the development of networks they found?

Reviewer #3 (Remarks to the Author):

This work proposes a novel approach to extract age-related brain networks by decomposing a dense matrix of age effects with ICA factorization. This approach is proposed based on a previous theory that functionally related areas mature together. This approach outperformed seed-based FC analysis and conventional group-ICA in detecting networks that better conform to anatomical boundaries. It's impressive seeing the spatially-distinct and GM-specific maturational networks. The main feedback provided here relates to clarification of methodology. It is also unclear if this is a methodological paper or a developmental paper. If developmental, then the developmental theory and interpretation is lacking.

1. The author did not provide information on how the netmat correlates with age. I guess they would be negatively correlated - similar to the group ICA, as they are quite co-localized. If so, it would be hard to conclude the matnet is better than group ICA in sensitivity. Pairs of the netmat and complementary networks are interesting. Could the author provide more interpretation on the biological meaning of the netmat-complementary network pair.

2. Regarding the decomposition of the maturational dense connectome, could the author provide more explanation on that? For conventional ICA conducted on voxel by timeseries matrix, I can understand that it is based on the assumption that the BOLD signal consists of a mixture of a bunch of relatively independent signal sources ("the network" in the context of brain). However, it's hard for me to understand the output of the ICA decomposition (factorization) on a connectome matrix. Is there any assumption on independent sources of the connectome matrices?

3. In addition, some modifications are needed to help readers to understand the new approach more easily. In Figure 1, "ICA factorization" should be shown next to the arrow from the maturational dense connectome (step 4) to the spatial map (step 5). Is the "spatial map" in figure 1 the same as the "spatial map" in figure 5? I got the number of matnets is 25 from the Methods, however, it will be helpful to know the number here. I've learned that the voxel within a spatial map has similar maturational profiles. But I'm still confused about "maturation profiles". Does it mean FC of voxels

within one component changes with age in a similar pattern? More interpretation of the "spatial map" is needed for readers in the "the framework" section.

4. Figure 3B (lower row). Can the author elaborate on how the map of age effect is calculated?

5. Is it possible to provide a figure showing components pairs side by side derived by the matnet factorization and group ICA?

6. In the split-half analysis, are the two halves age-matched? Was the whole analysis performed for each half respectively, including the met-net construction?

7. P8, L 281. The first sentence is confusing to me. Why "within the standard group-ICA +DR approach, the networks can be provided using "netmats. What does the "functional relationship" mean?

8. P8, L 286. The "maturational relationship" needs more clarification. If two matnets have a close "maturational relationship", does it mean the two matnets have similar age-related FC maps? I understand that each element of the maturational connectome indicates the "maturational relationship" of a pair of components and the maturation connectome is the basis of 3D embedding, so this clarification would be helpful for readers to understand what clusters in Figure 5 represent.

9. P8, L 288-292. Consider to add the corresponding metric's name here, for example: "As shown in Figure 5A, this involves regression of the maturational networks against columns of the thresholded maturational dense connectome in order to obtain an M components by N voxels matrix of estimated regression coefficients (an analogue of component timecourses of DR1 (b1 in Figure 5A), followed by calculation of M components by M components matrix, in which elements represent correlations between each pair of rows in the regression coefficient matrix (that is, the "maturational connectome")." Different letters, not all "N"s, should be used in representing different numbers of components and voxels (P8, L289). Also, could you provide the real number of M and N here?

10. Figure 5A, the matrix size (such as N voxels, M components) should be shown next to the matrix. It will be easier for readers to understand the matrix decomposition if the size is shown for b0, b1, maturational connectomes, etc.

11. P9 L298. How is the group number of "5" decided?

12. Figure 6B and supplementary Figure 12 show that the matnet-complement correlation is age-sensitive. It is not surprising as they are decomposed by a thresholded age-related t map. Was the "time course" for each map the mean time-course across voxels within the map? On the other hand, I am curious about how the time course of a matnet or a complement network relates to age. Will it show a negative correlation like group-ICA shows? Although mid- or long-range FC are shown in some the matnets, most of them are still regional.

13. P11 L 343. As far as I understand, the dense connectome is derived from the group-level regression model, as shown in figure 1. How did the author obtain "each subject's dense connectome"?

14. Discussion. Given the difference of subjects' brain size and brain structure relative to the 37-week template varies across ages, the registration accuracy may also change with age. is it possible that the maturational dense connectome can be related to registration accuracy?

15. Methods. Group-ICA and Maturational models: How the number of components of 25 decided? (P18, L639)

16. Maturation models of variation: what is "MIGP" ? (P18, L650)

17. Finally, the manuscript would benefit from closer evaluation of the developmental insights that this new method has brought to bear on this critical stage of human development.

Note to the Editor:

We have made changes to the analysis, reported in Figure 2B, on the relationship between the spatial distance and the similarity (i.e., spatial correlation) between pairs of seed-to-brain maps. Previously we reported a similar analysis using a small set of pre-selected seeds, which may not be convincing enough. In the revised version, we report the analysis based on the whole-brain cortical parcellation (300 clusters), thereby providing a more accurate evaluation of the above relationship. We changed the corresponding part of the manuscript (page 4, line 205) and added the description of the parcellation procedure to the Methods (page 22, line 784):

Furthermore, Figure 2B shows the relationship between the spatial distance and the similarity (i.e., spatial correlation) between 44850 pairs of seed-to-brain maps, computed following the parcellation of the cortex into 300 clusters. The relationship was strong for group-average correlation maps ($r = -.80$), which suggests that spatial distance may become a dominant factor for the fusion of the voxels into networks in analyses based on the correlational structure of the data, such as group-ICA. Conversely, the similarity between age-effect maps was more robust to the effect of distance between seeds used to produce these maps ($r = -.42$).

For the analysis of the relationship between similarity of seed-to-brain maps and the distance between them, cortical mask was parcellated into 300 clusters with k-means algorithm using voxel coordinates as input. The seed-to-brain group-average correlation and age-effect maps were calculated as above. Spatial distance between a pair of parcels was computed as a distance between their centres-of-gravity.

Reviewers' comments:**Reviewer #2 (Remarks to the Author):**

The work by Karolis and colleagues deals with the fundamental challenge of characterizing functional brain development before birth in humans. Despite some progress in this field, this remains an elusive research goal. While fetal fMRI has been the topic of a handful of studies before, this work may stand out for two reasons. First, they relied on a robust and unique dataset of 144 normally developing cases, acquired with a relatively long fMRI (compared to other fetal fMRI papers) sequence. Second, they did not spare the time to develop analytical methods that may efficiently deal with the problem of how to characterize resting state networks that 'appear from nothing' during development. While the focus is not always entirely clear (that is: biology or new methodology), a substantial part of the work describes the analytical framework to study and concludes that it is superior to the standard approaches. The analytical framework is convincing and new.

In the formulation of the analytical framework to study emergence of brain networks, the authors proposed two possibilities. First, by using dual regression – a technique that is well established in the resting-state fMRI community – in which gestational age is a covariate of interest in a multivariate regression model. Second, simply said - a new model where individual elements in the matrix are used in a multivariate regression model with age as covariate, and then combining these into a maturational network. The manuscript is well written, the findings of this work are supported by their data and methodology.

We thank the Reviewer for taking their time to provide a very accurate summary of our work.

The article could be improved if some of the biological underpinnings were better articulated. While the authors used numerous strategies to overcome fetal motion, it could be further disputed to what extent do these age-related findings originate from the many physiological changes during gestation and not brain functional maturation per se. More details about the last concern are below. The below list are my suggestions for improvements, which are mainly questions to be answered here or in the manuscript. Question number 1 might require more experiments to be done.

1. The authors introduce a new methodology and very convincingly deliver evidence that it is superior to an alternative approach (dual regression). However, they base their conclusion on a single dataset of fetal MRI, which is known to be extremely challenging, and many age-related phenomena, such as vasculature maturation or maternal physiology / movement, are very difficult to control for. The impact of the method on the field could be increased if the authors already utilized it on other data they might be in possession of, for example, on postnatal newborn, preterm or infant datasets. Could the authors please consider this suggestion or make an argument why the same purpose (emerging brain networks) cannot be tested on that dataset, too? Since there is a lot more published in postnatal field, even using EEG/MEG or other modalities, there are higher chances to validate the findings on dynamically changing network structures.

Thank you for the suggestion to apply the method to another dataset, which we agree would significantly increase the broader impact of the method. We have now also performed the matnet analysis using the data from the neonatal dHCP cohort, which encompasses newborn infants from 26 to 45 weeks (mean age 37.27 weeks). In this dataset matnets shows excellent agreement with group-ICA. Qualitatively, matnets tended to reveal features characteristic of more mature brains with a greater specificity, owing to the fact that our approach aims to describe “connectivity-in-making” as opposed to “connectivity-as-present”. The results and a new figure are now presented in a new section “Comparison of group-ICA and matnets in neonatal sample” (page 6, line 231) and are further discussed in the Discussion section (page 15, line 451):

We also tested the performance of the framework in the neonatal dHCP sample. Overall, matnets showed excellent agreement with group-ICA analysis of the same data. Furthermore, matnets revealed features characteristic of more mature brains with a greater specificity, such as more symmetrically distributed patterns across the two hemispheres and a nearly complete default mode network. Conceptually, a greater fractionation of group-ICA neonatal networks is not surprising, because compared to the “connectivity-as-present” representation ICA provides, matnets reconstruct maps of “connectivity-in-making”.

2. Fetal head motion patterns are also changing during gestation, for example by the decreasing space available for the fetus to move. I understand that they used the motion regressors during the signal processing, but was overall/total FD used in the developmental model as an additional covariate? This would perhaps help to counteract the GA-related changes in the movement patterns, which may mimic GA related changes in the functional network architecture.

We agree that head motion is an important factor in fetal fMRI, the amount of which can co-vary with age. However, we feel that caution is needed in deciding whether a metric such as total FD needs to be included as a covariate or not in the analysis. This is because its inclusion may in effect remove “true” age-effects due to its collinearity with age (for reasons correctly highlighted by the reviewer) and therefore have a detrimental effect on the power and accuracy of the analysis.

Unfortunately, there is no ground-truth test for distinguishing between “real” and a “trivial” (power-diminishing) effect of the variable representing overall motion. Consequently, we attempted to evaluate indirect evidence. For this, we considered how the landscape of age-related effects in the context interhemispheric connectivity between homologous left and right voxels changes as a result of inclusion of FD as a covariate. First, we found that the maps of age-effects with and without FD as a covariate are extremely collinear, $r = 0.98$, meaning that spatial configuration of the age-effect have not been affected. Second, we found that the differences between the two maps to a large degree commensurate with the strength of the age effect estimated without FD (i.e., spatial correlation spatial between age-effect t-map calculated without FD as a covariate and the difference between age-effect maps calculated with and without FD as a covariate: $r = -0.49$). In other words, there is a graded decrease of age effect with the inclusion of FD, whereby negatively values become less negative and vice versa for positively values. Finally, we note that the age effects demonstrate tissue specificity which is difficult to explain from the position that their occurrence is driven by the motion. Taking together, these observations have a simpler explanation based on the hypothesis of FD-age collinearity, whereas an alternative interpretation would presume a complex interaction between tissues, age and motion, for which we do not have substantial evidence.

We refer to the added materials in the method section (page 22, line 800):

We also considered whether we need to include a measure of a global motion (frame-wise displacement (FD)) as a covariate, but concluded against this (Supplementary Method 1).

and added Supplementary Methods 1 to the Supplementary Materials:

We evaluated whether an estimation of the age effect on the connectivity between pairs of voxels is confounded with the effect of the differences in the overall motion, given that the motion of older fetuses may be constrained by their own size and upside-down position. For this, a global measure of frame-wise displacement (FD) was calculated in the following steps. First, a mean of absolute FD for each motion parameter was calculated across time (altogether 96 values: 6 rotations & translations times 16 multiband stacks) in each subject. These means were collected into a 144 (number of subjects) x 96 matrix, which was then z-scored across rows (subjects). Finally, the first principal component was computed and used as a measure of global between-subject variation in motion.

We found that a small-effect correlation between age and FD, $r = -0.25$. Consequently, we analysed whether a potential confounding effect of FD alters age-effect statistics in a spatially varying manner, to which, unlike to a global effect, the ICA factorisation would be sensitive. An alternative hypothesis is that FD is not an independent factor but alters age-related statistics only because it is collinear with age. For this we considered age-related changes in interhemispheric connectivity between homologous left and right voxels.

First, we found that the maps of age-effect statistics computed with and without global FD as a covariate are highly correlated, $r = 0.98$. Furthermore, the inclusion of FD as a covariate resulted in a graded decrease of estimates of age-effect statistics with respect to the magnitude of the estimated age-effect (spatial correlation between age-effect t-map calculated without FD as a covariate and the difference between maps calculated with and without FD as a covariate: $r = -0.49$). In other words, the FD inclusion makes negatively values less negative and vice versa for positively values. Finally, we note the age effects tend to be tissue specific, i.e., tended to be more positive in the cortex and more negative in the white matter (Supplementary Figure 16), which is not expected if the source of association was motion. Taking together, the above observations can be explained based on the hypothesis of FD-age collinearity, whereas an alternative interpretation, presuming an independent effect of FD, entails a complex interaction between tissues, age and motion, for which we do not have substantial evidence. These considerations serve as a justification for not inclusion of FD in the downstream modelling.

3. On a similar note, other factors are changing as a function of gestational age. T2 time of the fetal brain is strongly correlated with GA (Counsell et al. AJNR Am J Neuroradiol. 2003 Sep; 24(8): 1654–1660), so for fetal fMRI studies, it might be worth using a variable TE in the range between 130-190 ms. (Blazejewska et al. Magn Reson Med. 2017 Sep;78(3):909-916.) The authors used 60, can they please comment on this? How would this affect the maturational networks?

The choice of the TE = 60 ms was dictated by the need to strike a balance between sampling frequency and the longer relaxation times of immature tissues; hence we chose a compromise with a significantly increased TE compared to adult-type acquisitions (e.g., TE = 33ms in HCP) but not so long that it would result in a marked loss of temporal resolution. We do not believe that using a variable TE would have been an appropriate solution to the

problem of maturational changes in tissue relaxation values, because it entails attempting to control for a range of confounding factors which would likely be even more problematic to address. One of the most obvious ones is that using a varying TE would result in a varying sampling rate, a factor which would significantly affect connectivity estimation at all levels of analyses, including the preprocessing stages such as denoising.

We would also like to highlight that the difference between the relaxation rate and TE is not as dramatic as suggested by the Reviewer, because the referred to recommendations in Blazejewski et al. are valid for 1.5T scanners, on which T2* times are much longer than for 3T scanners. The estimates most applicable to our sample are those obtained in neonates which are in the region of 100 ms (Ref #52). Furthermore, the same study (as well as others) indicate that the relationship between T2* and optimal TE is not straightforward, as a \sim TE=50 ms has been shown to be most sensitive for measuring a task effect.

However, we agree with the Reviewer that the effect of tissue composition on the T2* relaxation rate represents an intrinsic confound for which there is not a clear solution, and we now acknowledge this in Discussion.

The following parts were added to the main text (page 17, line 577):

The effect of tissue composition on the T2 relaxation rate may also represent an intrinsic confound for our analysis. The dHCP acquisition utilises a substantially longer TE (60 ms) compared to a benchmark adult acquisition (e.g., HCP protocol: 33 ms)⁵¹ in order to align with longer relaxation rates in the developing brain. However, assuming T2*=100 ms for neonates⁵¹ and that matching TE and T2* may (theoretically) provide higher SNR, the current TE may be a more “optimal” choice for the older fetuses, therefore, potentially biasing estimates of the age effects. However, this is not supported by the observed tendency of white matter seeds/voxels to show a negative association with age compared to the cortical regions, given that age-related tissue changes are likely to be more pronounced in the white matter than in the cortex, as white matter maturation occurs throughout gestation and myelination does not commence in most regions until the early neonatal period⁵². One would then expect greater positive age-related changes for the white matter if the effect was due to the SNR-TE relationship.*

4. Third, did the authors take into consideration the head position of the fetus? The positioning of the fetus towards the pelvic outlet towards the end of gestation might decrease the SNR, since the whole FOV is surrounded by more pelvic bones and more air filled bowels.

Thank you for raising this important point, which we now acknowledge as a limitation in the discussion, However, we also believe this factor is not critical to our results. In the worst case scenario, not controlling for this factor would diminish the power of matnet analysis, given that the latter leverages positive age-related associations and thus the older fetuses – in theory – would be the most “affected” ones.

We commented on this in the Discussion section (page 18, line 591) as follows:

Another potential confound is that fetuses tend to change position from pointing upwards to a head-down position later in the gestation, potentially affecting the signal. However, we do not believe this factor explains age-related increases in connectivity leveraged by the matnet analysis, as the head-down position would result in a decrease in SNR and consequently decreased estimates of connectivity strength, due to the effects of the surroundings such as the adjacent bones and air-filled bowels.

5. There have been prior works in fetal fMRI using ICA, one of the two methods the authors evaluate in the beginning, so the authors might want to consider comparing their results also with the networks reported in the (not yet cited) publications (e.g. Schopf et al. Int J Dev Neurosci. 2012 Feb ;30(1) :11-7., Schopf et

al. Front Hum Neurosci . 2014 Oct 2;8:775.), or dispute why their methodology may or may not reveal different networks than these publications.

Thank you for highlighting these pioneering studies. We would like to note that we are not in a position to dispute or support their results as they were derived using single-subject ICA and do not specifically involve modelling age-related changes during fetal development like our work. As our analyses are focused on group-level network estimation and characterization of development, we did not consider these papers as directly foundational for our study. However, we now cite them in order to acknowledge the pioneering status of their work (page 1, line 75, Refs 11 & 12).

Previous research has demonstrated that, despite enormous technological challenges, functional connectivity in utero can also be studied using resting-state fMRI⁸⁻¹².

6. The authors write “Therefore indicating that functional connections supporting high-level human cognition start emerging even prior to exposure to the extra utero environment.” It is not yet clear what the author’s hypotheses are for the biological underpinning of how higher level human cognition can be present, e.g. multisensory integration? They relate these findings mainly to past papers based on fMRI. However, can they elucidate what mechanism might be driving the establishment of such networks at the neuronal level? There are also other works that proposed that neural correlates of higher cognitive functions may emerge before birth, the authors might want use the findings of these previous works to support some of their hypotheses (e.g. Schopf et al. Front Hum Neurosci . 2014 Oct 2;8:775., Jakab Front Hum Neurosci. 2014 Oct 22;8:852.). Can the authors maybe explain these age related changes by some data on structural connectivity, e.g. to show when long-range association fibers emerge? How would regional changes in the resolution of the subplate and subplate-cortex connections affect the development of networks they found?

We apologise for the confusion. In this sentence from the abstract, we did not mean to imply that high-level cognition starts emerging during this time. Here, we refer only to the emergence of functional connectivity in the corresponding areas, which in itself may be driven by intrinsic factors. Discussing possible mechanisms would be highly speculative for this method-focused fMRI paper. Nevertheless, the finding that functional connectivity of high-order areas may develop in in utero are in line with recent evidence from a tissue staining study showing very early patterning of deep projection neurons in the frontal areas, which are traditionally considered to be a late maturing territory.

To avoid the misinterpretation in future we changed the corresponding sentence in the abstract (line 44) to:

indicating that functional connections of high-level associative areas start emerging even prior to exposure to the extra utero environment.

and referenced the above mentioned study (page 17, line 550):

They also align with a recent study showing an early patterning of deep projection neurons in the frontal lobe, which could provide a structural infrastructure for the functional connectivity of high-level associative areas⁴⁴

Reviewer #3 (Remarks to the Author):

This work proposes a novel approach to extract age-related brain networks by decomposing a dense matrix of age effects with ICA factorization. This approach is proposed based on a previous theory that functionally related areas mature together. This approach outperformed seed-based FC analysis and conventional group-ICA in detecting networks that better conform to anatomical boundaries. It’s impressive seeing the spatially-distinct and GM-specific maturational networks. The main feedback provided here relates to clarification of methodology. It is also unclear if this is a methodological paper or a developmental paper. If developmental, then the developmental theory and interpretation is lacking.

Thank you for your constructive comments which we hope we have fully addressed in the revision. The focus of the paper is methodological. This is emphasized with the addition of new analyses in a neonatal sample.

Nevertheless it remains biologically motivated. We clarify the nature of the paper in the introduction (page 2, line 96):

In this study, we hypothesised that a biologically-motivated analytical framework, that conceptualises functional brain network connectivity as a formative process, may provide a superior modelling alternative to the group-ICA for in-utero data.

1a. The author did not provide information on how the netmat correlates with age. I guess they would be negatively correlated - similar to the group ICA, as they are quite co-localized. If so, it would be hard to conclude the matnet is better than group ICA in sensitivity.

We believe the Reviewer refers here to the mass-univariate modelling of subject-specific maps obtained by propagating matnets through the standard pipeline, as described in Figure 1A. Whilst the Reviewer is absolutely correct in their prediction of negative associations, processing matnets via the standard pipeline would be conceptually inappropriate and therefore the conclusion that matnets are not better than group ICA in sensitivity is not accurate. This is because the matnets and associated complementary maps by design are a characterisation of the age-related emergence of connectivity; that is, matnets can be directly contrasted with the results of age-effect modelling of subject-specific maps derived using DR (of which examples are shown in Figure 5B, lower panel). Their propagation through the standard pipeline could not be interpreted in any meaningful way and the only result this would demonstrate is that the local negative age-related associations represent an intrinsic property of these data (which has little to do with the functional connectivity changes we attempt to infer). The section on univariate properties of the signal and result age-effect modelling for the group-ICA pipeline demonstrates this in depth already.

In the revised manuscript, we now emphasise that matnets are themselves the manifestations of age-related changes in functional connectivity (page 3, line 150):

[..] that is, we aim to derive spatial maps which themselves are the manifestations of age-related changes in functional connectivity.

1b Pairs of the netmat and complementary networks are interesting. Could the author provide more interpretation on the biological meaning of the netmat-complementary network pair.

In the revised manuscript, we discuss a duality between matnets and the complementary maps in detail. From a biological perspective the construct of complementary maps can be characterised as targets for the matnets' emerging connectivity. We clarify this in two places (page 10, line 349; and page 13, line 394):

From a biological perspective, matnets delineate areas which have similar targets for their emerging functional connections.

[..] this matrix represents a collection of complementary spatial maps, that depict targets to which corresponding matnets tend to develop connections to in an age-related manner, or to put it simply, the maps of the targets for their emerging connectivity.

2. Regarding the decomposition of the maturational dense connectome, could the author provide more explanation on that? For conventional ICA conducted on voxel by timeseries matrix, I can understand that it is based on the assumption that the BOLD signal consists of a mixture of a bunch of relatively independent signal sources ("the network" in the context of brain). However, it's hard for me to understand the output of the ICA decomposition (factorization) on a connectome matrix. Is there any assumption on independent sources of the connectome matrices?

We now clarify that the matnets can be viewed as independent "sources" of emerging connectivity and, consequently, the connectivity of any voxel in the brain conceptualised as a mixture of its connections with spatially independent matnets. The following changes were introduced (page 3, line 156; and page 10, line 350) :

An ICA factorisation of the maturational dense connectome is then performed to obtain spatially independent matnet maps, each of them associated with a characteristic profile of emerging connectivity.

Alternatively formulated, matnets can be viewed as independent “sources” of emerging connectivity, where their linear mixture determines age-related changes in connectivity of each voxel in the brain.

3.a In addition, some modifications are needed to help readers to understand the new approach more easily. In Figure 1, “ICA factorization” should be shown next to the arrow from the maturational dense connectome (step 4) to the spatial map (step 5).

Thank you for noticing this, we have now revised Figure 1B accordingly to address this comment.

3.b Is the “spatial map” in figure 1 the same as the “spatial map” in figure 5?

Yes, this is correct. We made changes to Figure 1B to clarify the relationship of the spatial maps referred to here and the subsequent results. Specifically, we have complementarily labeled them as “matnets”, to make clear that they correspond to the results reported in Figure 5 and Figure 6 (Figure 4 and 5 in the previous version of the manuscript).

3.c I got the number of matnets is 25 from the Methods, however, it will be helpful to know the number here.

We agree this would be helpful for the reader and have now stated this number in the caption to Figure 1

3.d I’ve learned that the voxel within a spatial map has similar maturational profiles. But I’m still confused about “maturation profiles”. Does it mean FC of voxels within one component changes with age in a similar pattern? More interpretation of the “spatial map” is needed for readers in the “the framework” section.

Yes, this is the correct interpretation by the Reviewer. We have now clarify this in the manuscript (page 3, line 156) as follows:

An ICA factorisation of the maturational dense connectome is then performed to obtain spatially independent matnet maps, each of them associated with a characteristic profile of emerging connectivity. In other words, as much as temporal correlations between voxels determines their participation in a particular group-ICA network, similarity in the age-related changes in connectivity between voxels determines their matnet participation.

4. Figure 3B (lower row). Can the author elaborate on how the map of age effect is calculated?

We have clarified this in the text and figure captions (currently Figure 4) by explicitly linking this result to the construct graphically demonstrated in Figure 1A.

In the text (page 8 line 289):

Meanwhile, the analysis of age-related changes in the spatial layout of the networks using the dual regression approach (mass-univariate modelling step in Figure 1A) [..]

In captions:

[..] and corresponding to the output of the mass-univariate modelling step in Figure 1A

5. Is it possible to provide a figure showing components pairs side by side derived by the matnet factorization and group ICA?

We added a new figure to Supplementary Materials (Supplementary Figure 12), where we paired group-ICA and matnets results using the Hungarian algorithm (Munkres, 1957) based on the spatial correlations between the two. The paired components are arranged in an ordered manner, starting from well matched (= high spatial correlation) to unmatched (= low spatial correlation). We now added this to the text (page 9 line 324):

A qualitative comparison to the paired group-ICA components (for the complete set - Supplementary Figure 12) demonstrates both the increased spatial specificity of the matnets approach and the differing sensitivity to interhemispheric and distal patterns of network participation.

6. In the split-half analysis, are the two halves age-matched? Was the whole analysis performed for each half respectively, including the met-net construction?

Thank you for requesting for this clarification, they were indeed age-matched. We have now clarified this in text (page 9, line 319):

In order to ascertain the robustness of the method, we repeated the analysis in approximately age-matched (mean age: 30.50 (3.23) and 30.42 (3.50), $t(142) = 0.14$, $p = .89$) split-half samples, computing matnets in each sample independently, and found a good replicability of the component spatial properties (Supplementary Figures 7-10).

7. P8, L 281. The first sentence is confusing to me. Why “within the standard group-ICA +DR approach, the networks can be provided using “netmats. What does the “functional relationship” mean?

We have now re-written this sentence using the more accurate term “functional connectivity” which we hope will help make the text clearer (page 11, line 362):

Firstly, a matrix of pairwise correlations between rows of the connectivity profile matrix summarises a similarity between matnets emerging connectivity profiles, in a similar way as a matrix of correlations between component timecourses outputted by DR1 (so called “netmats”³⁰) characterises functional connectivity between brain networks within the standard group-ICA+DR approach.

8. P8, L 286. The “maturational relationship” needs more clarification. If two matnets have a close “maturational relationship”, does it mean the two matnets have similar age-related FC maps? I understand that each element of the maturational connectome indicates the “maturational relationship” of a pair of components and the maturation connectome is the basis of 3D embedding, so this clarification would be helpful for readers to understand what clusters in Figure 5 represent.

The Reviewer provides an exact definition of the meaning of “maturational relationships” in this context. As suggested, we have now changed the text using a more precise terminology, substituting “maturational relationships” with “similarity between emerging connectivity profiles” where appropriate, e.g. (page 11 line 362 and page 11 line 371):

[..] a matrix of pairwise correlations between rows of the connectivity profile matrix summarises a similarity between matnets emerging connectivity profiles

[..] with a shorter distance between networks being indicative of a greater similarity between their emerging connectivity profiles [..]

9. P8, L 288-292. Consider to add the corresponding metric’s name here, for example: “As shown in Figure 5A, this involves regression of the maturational networks against columns of the thresholded maturational dense connectome in order to obtain an M components by N voxels matrix of estimated

regression coefficients (an analogue of component timecourses of DR1 (b_1 in Figure 5A), followed by calculation of M components by M components matrix, in which elements represent correlations between each pair of rows in the regression coefficient matrix (that is, the “maturational connectome”).” Different letters, not all “N”s, should be used in representing different numbers of components and voxels (P8, L289). Also, could you provide the real number of M and N here?

We have now changed this as requested and added the real number of M and N to the text (page 11 line 356):

In an analogy to the computation of component temporal courses in the standard-approach using the DR1 step (Figure 1A), emerging connectivity profiles associated with matnets are computed as a matrix (here $M = 25$ components by $N = 53\,443$ voxels) of regression slope coefficients by regressing matnet maps against columns of the thresholded maturational dense connectome (Figure 6A).

10. Figure 5A, the matrix size (such as N voxels, M components) should be shown next to the matrix. It will be easier for readers to understand the matrix decomposition if the size is shown for b_0 , b_1 , maturational connectomes, etc.

We have now added this information as requested in the figure 6A (formerly, Figure 5A).

11. P9 L298. How is the group number of “5” decided?

Thank you for requesting this clarification, which we agree would be helpful for the reader to understand. The choice of 5 groups was made by identifying the most refined partitioning which did not produce single-component groups. We have now added this to the manuscript (page 11, line 373):

In further analyses we used a 5-group partitioning which was the finest partitioning that does not produce single-network groups.

12. Figure 6B and supplementary Figure 12 show that the matnet-complement correlation is age-sensitive. It is not surprising as they are decomposed by a thresholded age-related t map.

Thank you for highlighting this point. Thresholding eliminates a regional differentiation in age-negative associations, but there is still a non-negligible probability that the decomposition, via complex multivariate interactions, may still result in pairs in which connectivity shows a negative association with age. However, as we note, the analysis primarily demonstrates a route to analyse individual variability and identify inter-regional differences in maturational trajectories, as shown by the differences in the best fitting function in Figure 7B (formerly, Figure 6B). We now emphasise the inter-areal differences in the trajectories in the following text (page 13, line 424):

Figure 7B shows the results for two maturational groups, with differing age-related trajectories, whereas age-related trajectories for all 25 components are shown in Supplementary Figure 13.

Was the “time course” for each map the mean time-course across voxels within the map?

Apologies for not being clear on this. This information was embedded in the figure caption but would have been far clearer in the main text of the manuscript. We have now moved it to the main text to ensure transparency (page 13, line 420):

Here, we estimated the temporal coupling between matnets and their complementary maps as a function of age. For this, both the matnets and their complementary maps were thresholded at $z > 5$ in order to reduce a degree of potential spatial overlap between the two and their time courses were computed as weighted averages of the above-threshold voxels.

On the other hand, I am curious about how the time course of a matnet or a complement network relates to age. Will it show a negative correlation like group-ICA shows? Although mid- or long-range FC are shown in some the matnets, most of them are still regional.

Please refer to our answer to question 1a.

13. P11 L 343. As far as I understand, the dense connectome is derived from the group-level regression model, as shown in figure 1. How did the author obtain “each subject’s dense connectome”?

The dense connectome consists of the correlations between each pair of voxels in each subject, as shown in Figure 1B. In the revised manuscript we no longer report this analysis (neither in the main text nor in Supplementary Materials) because it is not essential for the study (in the previous version the results were shown in Supplementary Material only and have never been discussed) and also because a large amount of additional material has been incorporated during the current revision.

14. Discussion. Given the difference of subjects’ brain size and brain structure relative to the 37-week template varies across ages, the registration accuracy may also change with age. Is it possible that the maturational dense connectome can be related to registration accuracy?

We agree with the Reviewer that this is a fundamental issue, that cannot never be completely resolved by nature of the changing fetal brain. We used a very comprehensive approach to the group-space registration (graphically depicted in Supplementary Figure 15A) to ameliorate this issue, as we have previously done for neonatal data (Ref #6). The core of this approach is to utilise the temporal smoothness of the spatio-temporal template (constructed using Gaussian kernel regression) and sequential template-to-template alignments. The non-linear warps are then concatenated with within-subject mapping between fMRI and structural data and with structural-to-template mapping into a single warp that maps the data from the native functional space into a group space. This approach avoids a necessity of computing large (error-prone) deformations, thereby achieving a remarkable alignment even for morphologically distant brains. We now show an example of warping GA25 template into the GA37 template (Supplementary Figure 15B) to highlight this point. In general, we expect that inaccuracy in registration will to some degree be balanced out between ages by diverging factors: in younger a cause of misregistration is likely to be a simple brain morphology that lacks distinct landmarks; in older fetuses it is the unique complexity of gyrified brain that makes it difficult to fit a standard space.

We have added the following to the text (page 18 line 598):

Finally, the registration accuracy represents a fundamental issue, that can never be completely resolved by nature of the changing fetal brain. To ameliorate this issue, we used a very comprehensive approach to the group-space registration, previously exploited for the neonatal data⁶, which avoids a necessity of computing large – and error-prone – deformations and at the same time achieving a remarkable alignment even for morphologically distant brains (see Methods for the description). In general, we expect that inaccuracy in registration will to some degree be balanced out between ages by diverging factors: in younger a cause of misregistration is likely to be a simple brain morphology that lacks distinct landmarks; in older fetuses it is the unique complexity of gyrified brain that makes it difficult to fit a standard space. However, further work is needed to fully assess the effects of the template choices and registration procedures for this challenging type of data.

15. Methods. Group-ICA and Maturational models: How the number of components of 25 decided? (P18, L639)

We used the dimensionality that was previously used in neonates (Ref #6). We clarify this in the captions for Figure 1 and in the text (page 22 line 793):

The number of derived components was set to 25, in accordance with the published research in neonates⁶

16. Maturational models of variation: what is “MIGP” ? (P18, L650)

As this term was defined in the previous section of the manuscript, here we refer to its definition more explicitly (page 22 line 807)

[..] analogous to the MIGP¹⁴ step of the group-ICA [..]

17. Finally, the manuscript would benefit from closer evaluation of the developmental insights that this new method has brought to bear on this critical stage of human development.

Thank you for this important suggestion. We have complemented a detailed description of the macroscopic patterns of connectivity that our methods help to uncover via the analysis of maturational connectome and maturational hubs with the discussion of the results in neonates (page 15, line 453):

Furthermore, matnets revealed features characteristic of more mature brains with a greater specificity, such as more symmetrically distributed patterns across the two hemispheres and a nearly complete default mode network. Conceptually, a greater fractionation of group-ICA neonatal networks is not surprising, because compared to the “connectivity-as-present” representation ICA provides, matnets reconstruct maps of “connectivity-in-making”.

With respect to the fetal sample, we called the attention of the reader to the relevant passage in the revised manuscript by prefacing it with the following (page 15, line 489):

An additional level of insight into the developmental sequelae of the fetal functional brain and the shaping of future network architecture is provided by considering matnets in association with their complementary maps, with the latter characterising the matnets’ emerging connectivity profiles. This leads towards two novel constructs: the maturational connectome, that summarises similarity of emerging connectivity profiles between pairs of matnets (Figure 6), and maturational hubs, that represent common targets for the matnets’s maturing connections (Figure 7). Together, their analyses allow us to characterise macroscopic patterns of connectivity that emerge during this critical stage of human development.

We also highlighted findings that show an agreement with the existing literature in neonates and post-mortem fetal samples (page 17 line 550; and page 17 line 556):

They also align with a recent study showing an early patterning of deep projection neurons in the frontal lobe, which could provide a structural infrastructure for the functional connectivity of high-level associative areas⁴⁴.

Similarly, consistent with previous findings in preterm neonates⁴, no strong evidence of the links between medial posterior and anterior nodes of the default mode network were observed, which at this period appear to be integrated, respectively, within parietal and frontal networks and lack a prominent role in the observed whole brain functional architecture. Interestingly, a nearly complete default mode network identified with matnets in neonates contained only a small cluster of voxels in the medial parietal cortex, supporting the evidence that a fully functional DMN may emerge only as late as at the age of 3⁴⁵.

REVIEWERS' COMMENTS:

Reviewer #2 (Remarks to the Author):

The authors have made significant revisions to their original submission, including conducting an evaluation of their proposed method on a larger dataset (dHCP newborns), to reveal the emergence of functional brain networks. This has greatly enhanced the generalizability of their approach.

I strongly recommend that the authors make their code or data available whenever possible.

The current form of the manuscript represents a significant contribution to the field and is worthy of acceptance based on its scientific merits.

Reviewer #3 (Remarks to the Author):

The authors have responded fully to my critiques. I recommend this paper for publication.

Reviewer #4 (Remarks to the Author):

The authors have responded fully to my critiques. I recommend this paper for publication.